# ACTIVATION NOISE FOR EBM CLASSIFICATION

## ABSTRACT

We study activation noise in a generative energy-based modeling setting during training for the purpose of regularization. We prove that activation noise is a general form of dropout. Then, we analyze the role of activation noise at inference time and demonstrate it to be utilizing sampling. Thanks to the activation noise we observe about 200% improvement in performance (classification accuracy). Later, we not only discover, but also prove that the best performance is achieved when the activation noise follows the same distribution both during training and inference. To explicate this phenomenon, we provide theoretical results that illuminate the roles of activation noise during training, inference, and their mutual influence on the performance. To further confirm our theoretical results, we conduct experiments for five datasets and seven distributions of activation noise.

## 1 INTRODUCTION

Whether it is for performing regularization (Moradi et al., 2020) to mitigate overfitting (Kukacka et al., 2017) or for ameliorating the saturation behavior of the activation functions, thereby aiding the optimization procedure (Gulcehre et al., 2016), injecting noise to the activation function of neural networks has been shown effective (Xu et al., 2012). Such activation noise denoted by $z$, is added to the output of each neuron of the network (Tian & Zhang, 2022) as follows:

$$t = s + z = f(\sum_i w_i x_i + b) + \alpha \bar{z} \qquad (1)$$

where $w_i$, $x_i$, $b$, $s$, $f(\cdot)$, $\alpha$, $\bar{z}$, and $t$ stand for the $i$th element of weights, the $i$th element of input signal, the bias, the raw/un-noisy activation signal, the activation function, the noise scalar, the normalized noise (divorced from its scalar $\alpha$) originating from any distribution, and the noisy output, respectively. Studying this setting inflicted by noisy units is of significance, because it resembles how neurons of the brain in the presence of noise learn and perform inference (Wu et al., 2001).

In the literature, training with input/activation noise has been shown to be equivalent to loss regularization: a well-studied regularization scheme in which an extra penalty term is appended to the loss function. Also, injecting noise has been shown to keep the weights of the neural network small, which is reminiscent of other practices of regularization that directly limit the range of weights (Bishop, 1995). Furthermore, injecting noise to input samples (or activation functions) is an instance of data augmentation (Goodfellow et al., 2016). Injecting noise practically expands the size of the training dataset, because each time training samples are exposed to the model, random noise is added to the input/latent variables rendering them different every time they are fed to the model. Noisy samples therefore can be deemed as new samples which are drawn from the domain in the vicinity of the known samples: they make the structure of the input space smooth, thereby mitigating the curse of dimensionality and its consequent patchiness/sparsity of the datasets. This smoothing makes it easier for the neural network to learn the mapping function (Vincent et al., 2010).

In the existing works, however, the impacts of activation noise have been neither fully understood during training time nor broached at the inference time, not to mention the lack of study on the relationship between having activation noise at training and inference, especially for the generative energy-based modeling (EBM). In this paper, we study those issues: for the EBM setting, for the first time, we study the empirical and theoretical aspects of activation noise not only during training time but also at inference and discuss how these two roles relate to each other. We prove that, during training, activation noise (Gulcehre et al., 2016) is a general form of dropout (Srivastava et al., 2014). This is interesting because dropout has been widely adopted as a regularization scheme. We then

formulate and discuss the relationship between activation noise and two other key regularization schemes: loss regularization and data augmentation.

We also prove that, during inference, adopting activation noise can be interpreted as sampling the neural network. Accordingly, with activation noise during inference, we estimate the energy of the EBM. Surprisingly, we discover that there is a very strong interrelation between the distribution of activation noise during training and inference: the performance is optimized when those two follow the same distributions. We also prove how to find the distribution of the noise during inference to minimize the inference error, thereby maximizing the performance as high as 200%. Overall, our main contributions in this paper are as follows:

- We prove that, during training, activation noise is a general form of dropout. Afterward, we establish the connections between activation noise and loss regularization/data augmentation. With activation noise during inference as well as training, we observe about 200% improvement in performance (classification accuracy), which is unprecedented. Also, we discover/prove that the performance is maximized when the noise in activation functions follow the same distribution during both training and inference.

- To explain this phenomenon, we provide theoretical results that illuminate the two strikingly distinct roles of activation noise during training and inference. We later discuss their mutual influence on the performance. To examine our theoretical results, we provide extensive experiments for five datasets, many noise distributions, various noise values for the noise scalar $\alpha$, and different number of samples.

## 2    RELATED WORKS

Our study touches upon multiple domains: (i) neuroscience, (ii) regularization in machine learning, (iii) generative energy-based modeling, and (iv) anomaly detection and one-class classification.

(i) Studying the impact of noise in artificial neural networks (ANNs) can aid neuroscience to understand the brain's operation (Lindsay, 2020; Richards et al., 2019). From neuroscience, we know that neurons of the brain (as formulated by Eq. 1) never produce the same output twice even when the same stimuli are presented because of their internal noisy biological processes (Ruda et al., 2020; Wu et al., 2001; Romo et al., 2003). Having a noisy population of neurons if anything seems like a disadvantage (Averbeck et al., 2006; Abbott & Dayan, 1999); then, how does the brain thwart the inevitable and omnipresent noise (Dan et al., 1998)? We provide new results on top of current evidence that noise can indeed enhance both the training (via regularization) and inference (by error minimization) (Zylberberg et al., 2016; Zohary et al., 1994).

(ii) Injecting noise to neural networks is known to be a regularization scheme: regularization is broadly defined as any modification made to a learning algorithm that is intended to mitigate overfitting: reducing the generalization error but not its training error (Kukacka et al., 2017). Regularization schemes often seek to reduce overfitting (reduce generalization error) by keeping weights of neural networks small (Xu et al., 2012). Hence, the simplest and most common regularization is to append a penalty to the loss function which increases in proportion to the size of the weights of the model. However, regularization schemes are diverse (Moradi et al., 2020); in the following, we review the popular regularization schemes: weight regularization (weight decay) (Gitman & Ginsburg, 2017) penalizes the model during training based on the magnitude of the weights (Van Laarhoven, 2017). This encourages the model to map the inputs to the outputs of the training dataset such that the weights of the model are kept small (Salimans & Kingma, 2016). Batch-normalization regularizes the network by reducing the internal covariate shift: it scales the output of the layer, by standardizing the activations of each input variable per mini-batch (Ioffe & Szegedy, 2015).

Ensemble learning (Zhou, 2021) trains multiple models (with heterogeneous architectures) and averages the predictions of all of them (Breiman, 1996). Activity regularization (Kilinc & Uysal, 2018b) penalizes the model during training based on the magnitude of the activations (Deng et al., 2019; Kilinc & Uysal, 2018a). Weight constraint limits the magnitude of weights to be within a range (Srebro & Shraibman, 2005). Dropout (Srivastava et al., 2014) probabilistically removes inputs during training: dropout relies on the rationale of ensemble learning that trains multiple models. However, training and maintaining multiple models in parallel inflicts heavy computational/memory expenses. Alternatively, dropout proposes that a single model can be leveraged to simulate training an expo-

nential number of different network architectures concurrently by randomly dropping out nodes during training (Goodfellow et al., 2016). Early stopping (Yao et al., 2007) monitors the model's performance on a validation set and stops training when performance starts to degrade (Goodfellow et al., 2016). Data augmentation, arguably the best regularization scheme, creates fake data and augments the training set (Hernandez-Garcia & Konig, 2018). Label smoothing (Lukasik et al., 2020) is commonly used in training deep learning (DL) models, where one-hot training labels are mixed with uniform label vectors (Meister et al., 2020). Smoothing (Xu et al., 2020) has been shown to improve both predictive performance and model calibration (Li et al., 2020b; Yuan et al., 2020).

Noise schemes inject (usually Gaussian) noise to various components of the machine learning (ML) systems: activations, weights, gradients, and outputs (targets/labels) (Poole et al., 2014). In that, noise schemes provide a more generic and therefore more applicable approach to regularization that is invariant to the architectures, losses, activations of the ML systems, and even the type of problem at hand to be addressed (Holmstrom & Koistinen, 1992). As such, noise has been shown effective for generalization as well as robustness of a variety of ML systems (Neelakantan et al., 2015).

(iii) Our simulation setting in this paper follows that of Generative EBM (we briefly say EBM henceforth) (LeCun et al., 2006). EBM (Nijkamp et al., 2020) is a class of maximum likelihood model that maps each input to an un-normalized scalar value named energy. EBM is a powerful model that has been applied to many different domains, such as structured prediction (Belanger & McCallum, 2016), machine translation (Tu et al., 2020), text generation (Deng et al., 2020), reinforcement learning (Haarnoja et al., 2017), image generation (Xie et al., 2016), memory modeling (Bartunov et al., 2019), classification (Grathwohl et al., 2019), continual learning (Li et al., 2020a), and biologically-plausible training (Scellier & Bengio, 2017). (iv) Our EBM setting leverages separate autoencoders (Chen et al., 2018) for classification, in that it resembles anomaly detection scenarios (Zhou & Paffenroth, 2017; An & Cho, 2015) and also one-class classification (Ruff et al., 2018; Liznerski et al., 2020; Perera & Patel, 2019; Sohn et al., 2020).

## 3 EBM AND ACTIVATION NOISE DURING TRAINING AND INFERENCE

EBM is a class of maximum likelihood model that determines the likelihood of a data point $\boldsymbol{x} \in \mathcal{X} \subseteq \mathbb{R}^D$ using the Boltzmann distribution:

$$p_{\boldsymbol{\theta}}(\boldsymbol{x}) = \frac{\exp(-E_{\boldsymbol{\theta}}(\boldsymbol{x}))}{\Omega(\boldsymbol{\theta})}, \quad \Omega(\boldsymbol{\theta}) = \int_{\boldsymbol{x} \in \mathcal{X}} \exp(-E_{\boldsymbol{\theta}}(\boldsymbol{x}))d\boldsymbol{x} \tag{2}$$

where $E_{\boldsymbol{\theta}}(\boldsymbol{x}) : \mathbb{R}^D \to \mathbb{R}$, known as the energy function, is a neural network parameterized by $\boldsymbol{\theta}$, that maps each data point $\boldsymbol{x}$ to a scalar energy value, and $\Omega(\boldsymbol{\theta})$ is the partition function. To solve the classification task in EBM utilizing activation noise, we adjust the general formulation of EBM in Eq. 2 as follows: given a class label $y$ in a discrete set $\mathcal{Y}$ and the activation noise $\boldsymbol{z}$ during training, for each input $\boldsymbol{x}$, we use the Boltzmann distribution to define the conditional likelihood as follows:

$$p_{\boldsymbol{\theta}}(\boldsymbol{x} \mid y, \boldsymbol{z}) = \frac{\exp\left(-E_{\boldsymbol{\theta}}(\boldsymbol{x} \mid y, \boldsymbol{z})\right)}{\Omega(\boldsymbol{\theta} \mid y, \boldsymbol{z})}, \quad \Omega(\boldsymbol{\theta} \mid y, \boldsymbol{z}) = \int_{\boldsymbol{x} \in \mathcal{X}} \exp\left(-E_{\boldsymbol{\theta}}\left(\boldsymbol{x} \mid y, \boldsymbol{z}\right)\right) d\boldsymbol{x} \tag{3}$$

where $E_{\boldsymbol{\theta}}(\boldsymbol{x} \mid y, \boldsymbol{z}) : (\mathbb{R}^D, \mathbb{N}, \mathbb{R}^F) \to \mathbb{R}$ is the energy function that maps an input given a label and noise to a scalar energy value $E_{\boldsymbol{\theta}}(\boldsymbol{x} \mid y, \boldsymbol{z})$, and $\Omega(\boldsymbol{\theta} \mid y, \boldsymbol{z})$ is the normalization function.

### 3.1 TRAINING WITH ACTIVATION NOISE

During training, we want the distribution defined by $E_{\boldsymbol{\theta}}$ to model the data distribution $p_D(\boldsymbol{x}, y)$, which we achieve by minimizing the negative log likelihood $\mathcal{L}_{\mathrm{ML}}(\boldsymbol{\theta}, q(\boldsymbol{z}))$ of the data as follows:

$$(\boldsymbol{\theta}^*, q^*(\boldsymbol{z})) = \underset{\boldsymbol{\theta}, q(\boldsymbol{z})}{\arg\min} \, \mathcal{L}_{\mathrm{ML}}(\boldsymbol{\theta}, q(\boldsymbol{z})), \quad \mathcal{L}_{\mathrm{ML}}(\boldsymbol{\theta}, q(\boldsymbol{z})) = \mathbb{E}_{(\boldsymbol{x},y)\sim p_D; \, \boldsymbol{z}\sim q(\boldsymbol{z})} \left[-\log p_{\boldsymbol{\theta}}(\boldsymbol{x} \mid y, \boldsymbol{z})\right] \tag{4}$$

where $q(\boldsymbol{z})$ is the distribution of the activation noise $\boldsymbol{z}$ during training.

We explicate the relationships between activation noise during training and (i) dropout, (ii) loss regularization, and (iii) data augmentation. We start by presenting theoretical results illuminating that activation noise is a general form of dropout. For that, we define the *negate* distribution of a

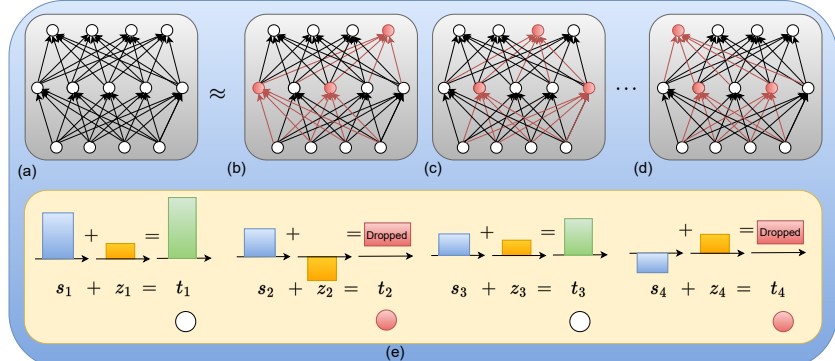

Figure 1: To mitigate overfitting, dropout proposes that a single model can simulate an exponential number of different network architectures concurrently by randomly dropping out nodes during training. As visualized, the neural network in Fig. (a) approximates many neural sub-networks such as those in Figs. (b), (c), and (d) where dropped neurons are painted red. Via this *virtual ensemble learning*, dropout mitigates overfitting. In this paper, we prove that activation noise generalizes dropout. As shown in Fig. (e), for activation noise at each neuron, the activation signal $s_i$ (shown with blue) is added by a noise $z_i$ (shown with yellow) which could come from a variety of distributions. When the noise distribution is the negate of the signal distribution for a neuron, that neuron is dropped (shown with red color), and in this case, the activation noise *reduces* to dropout.

*primary* distribution, based on which we derive Activation Noise Generality Proposition, stating that dropout is a special case of activation noise.

**Definition 1** (Negate Random Variable)**:** *We define random variable $W : \boldsymbol{\Sigma} \to \boldsymbol{E}$ as the negate of random variable $X : \boldsymbol{\Delta} \to \boldsymbol{F}$, denoted by $X \nmid W$ (where $\boldsymbol{\Sigma}, \boldsymbol{\Delta}, \boldsymbol{E}, \boldsymbol{F} \subset \mathbb{R}$) if the outcome of $W$ negates the outcome of $X$; mathematically speaking, $x + w = 0$.*

**Proposition 1** (Activation Noise Generality Proposition)**:** *When the noise at a given neuron comes from the negate random variable $Z$ with respect to the signal $S$, i.e., $Z \nmid S$, the activity noise drops out (the signal of) that neuron. Specifically, the summation of signal and noise becomes zero, $s + z = 0$ for all outcomes.*

*Proof:* The proof follows from the definition of the negate random variable.

This theoretical result implies that activation noise can be considered as a general form of dropout: Fig. 1 visualizes how the activation noise reduces to dropout. In Fig. 2 we compare the performances of activation noise with dropout with the simulation setting as for Fig. 3.

Now we explain how activation noise relates to loss regularization and data augmentation. To that end, we consider the EBM setting leveraging multiple distinct autoencoders for classification: one autoencoder is used for each class. We first write the empirical error without noise in the form of the mean squared error (MSE) loss function as follows:

$$I_{\boldsymbol{\theta}}^s = \sum_{y \in \mathcal{Y}} \int_{\boldsymbol{x}} \|f_{\boldsymbol{\theta}}(\boldsymbol{x} \mid y) - \boldsymbol{x}\|^2 p_D(\boldsymbol{x}, y) d\boldsymbol{x} = \sum_{y \in \mathcal{Y}} \sum_k \int_{x_k} \left[ f_{\boldsymbol{\theta}}^k(\boldsymbol{x} \mid y) - x_k \right]^2 p_D(\boldsymbol{x}, y) dx_k \quad (5)$$

where $f_{\boldsymbol{\theta}}(\boldsymbol{x} \mid y)$ denotes the model parameterized by $\boldsymbol{\theta}$ given label $y$, and the energy is determined by $E_{\boldsymbol{\theta}}(\boldsymbol{x} \mid y) = \|f_{\boldsymbol{\theta}}(\boldsymbol{x} \mid y) - \boldsymbol{x}\|^2$. Meanwhile, $f_{\boldsymbol{\theta}}^k(\boldsymbol{x} \mid y)$ and $x_k$ refer to the $k$th element of the output of the model and desired target (i.e., the original input), respectively. With activation noise, however, we have

$$I_{\boldsymbol{\theta}} = \sum_{y \in \mathcal{Y}} \sum_k \int_{\boldsymbol{z}} \int_{x_k} \left[ f_{\boldsymbol{\theta}}^k(\boldsymbol{x} \mid y, \boldsymbol{z}) - x_k \right]^2 p_D(\boldsymbol{x}, y) q(\boldsymbol{z}) dx_k d\boldsymbol{z} \quad (6)$$

where $f_{\boldsymbol{\theta}}^k(\boldsymbol{x} \mid y, \boldsymbol{z})$ denotes the $k$th element of the output of the noisy model. The noise $\boldsymbol{z}$ comes from distribution $q(\boldsymbol{z})$ during training. Expanding the network function $f_{\boldsymbol{\theta}}^k(\boldsymbol{x} \mid y, \boldsymbol{z})$ into the signal response $f_{\boldsymbol{\theta}}^k(\boldsymbol{x} \mid y)$ and the noisy response $h_{\boldsymbol{\theta}}^k(\boldsymbol{x} \mid y, \boldsymbol{z})$ to give

$$f_{\boldsymbol{\theta}}^k(\boldsymbol{x} \mid y, \boldsymbol{z}) = f_{\boldsymbol{\theta}}^k(\boldsymbol{x} \mid y) + h_{\boldsymbol{\theta}}^k(\boldsymbol{x} \mid y, \boldsymbol{z}), \quad (7)$$

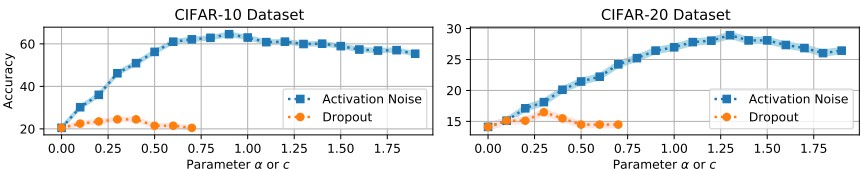

Figure 2: Comparison between activation noise and dropout for CIFAR-10 and CIFAR-20 datasets as the value of noise scalar $\alpha$ and dropout scalar $c$ increase (for $n = 1000$).

we can write Eq. 6 as $I_{\boldsymbol{\theta}} = I_{\boldsymbol{\theta}}^s + I_{\boldsymbol{\theta}}^z$, where $I_{\boldsymbol{\theta}}^z$ encapsulates the noisy portion of the loss given by

$$I_{\boldsymbol{\theta}}^z = \sum_{y \in \mathcal{Y}} \sum_k \int_{\boldsymbol{z}} \int_{x_k} \left[ h_{\boldsymbol{\theta}}^k(\boldsymbol{x} \mid y, \boldsymbol{z})^2 + 2 h_{\boldsymbol{\theta}}^k(\boldsymbol{x} \mid y, \boldsymbol{z})(f_{\boldsymbol{\theta}}^k(\boldsymbol{x} \mid y) - x_k) \right] p_D(\boldsymbol{x}, y) q(\boldsymbol{z}) dx_k d\boldsymbol{z}. \quad (8)$$

The term $I_{\boldsymbol{\theta}}^z$ can be viewed as the loss regularizer, specifically Tikhonov regularizer, as has been shown in (Bishop, 1995). In general, not just for MSE as we have done, but for any arbitrary loss function $v(f_{\boldsymbol{\theta}}(\boldsymbol{x} \mid y, \boldsymbol{z}), \boldsymbol{x})$, we can re-write it as a combination of the losses pertaining to the signal component plus the noise component and then the same result would hold. As suggested before, the second term of error, besides being a regularizer, can also be seen as the loss for an auxiliary dataset that is interleaved with the primary dataset during training. Hence, this way it can be said that the noise augments the dataset (Goodfellow et al., 2016).

This section concerned how activation noise relates to dropout, loss regularization, and data augmentation. As alluded to, activation noise during training is beneficial for the performance; but, what if we used activation noise also during inference? We will now answer this question: for EBM, activation noise can be also beneficial in inference. Specifically, we will first present the role of activation noise during inference in the EBM framework and then present the experiments yielding surprising results demonstrating the effectiveness of activation noise during inference.

### 3.2 INFERENCE WITH ACTIVATION NOISE

Consider the inference phase assuming training has been done (i.e., when $\boldsymbol{\theta}^*$ has been determined). Given a test data point $\boldsymbol{x}$, we *estimate* the energy of our trained model $E_{\boldsymbol{\theta}^*}(\boldsymbol{x} \mid y, \boldsymbol{z})$ with many different noise realizations $\boldsymbol{z}$ following *inference* noise distribution $r(\boldsymbol{z})$ which are averaged over to produce the energy. Therefore, the noise distribution $r(\boldsymbol{z})$ at the *inference* can be considered as a sampler. Probabilistically speaking, we measure the expectation of the energy $E_{\boldsymbol{\theta}^*}(\boldsymbol{x} \mid y, \boldsymbol{z})$ with respect to distribution $r(\boldsymbol{z})$ as follows:

$$\bar{E}_{\boldsymbol{\theta}^*}(\boldsymbol{x} \mid y) = \mathbb{E}_{\boldsymbol{z}}\left[E_{\boldsymbol{\theta}^*}(\boldsymbol{x} \mid y, \boldsymbol{z})\right] = \int_{\boldsymbol{z}} E_{\boldsymbol{\theta}^*}(\boldsymbol{x} \mid y, \boldsymbol{z}) r(\boldsymbol{z}) d\boldsymbol{z}. \quad (9)$$

The variance of $E_{\boldsymbol{\theta}^*}(\boldsymbol{x} \mid y, \boldsymbol{z})$, is determined by $\sigma^2 = \int_{\boldsymbol{z}} E_{\boldsymbol{\theta}^*}(\boldsymbol{x} \mid y, \boldsymbol{z})^2 r(\boldsymbol{z}) d\boldsymbol{z} - \bar{E}_{\boldsymbol{\theta}^*}(\boldsymbol{x} \mid y)^2$. In practice, because the calculation of the integral in Eq. 9 is intractable, we perform the inference via $\hat{E}_{\boldsymbol{\theta}^*}(\boldsymbol{x} \mid y) = \frac{1}{n} \sum_{i=1}^n E_{\boldsymbol{\theta}^*}(\boldsymbol{x} \mid y, \boldsymbol{z}^{(i)})$ where the integral in Eq. 9 is numerically approximated via sampling from the distribution $r(\boldsymbol{z})$ which generates the noise samples $\{\boldsymbol{z}^{(i)}\}$. Finally, given an input $\boldsymbol{x}$, the class label predicted by our EBM is the class with the smallest energy at $\boldsymbol{x}$, we find the target class via $\hat{y} = \arg\min_{y' \in \mathcal{Y}} \hat{E}_{\boldsymbol{\theta}^*}(\boldsymbol{x} \mid y')$.

This approach of classification is a common formulation for making inference which is derived from Bayes' rule. There is one difference, however, and that is in EBM classification we seek the class whose energy is the minimum as the class that the input data belongs to.

In the end note that, as discussed in this section, activation noise not only generalizes dropout during training, as a regularization scheme, but also offers the opportunity of sampling the model during inference (to minimize the inference error) possibly using a wide range of noise distributions as the sampler; this is advantageous to dropout that is only applicable during the training. In the next section, we will first present the simulation settings detailing the architectures used to conduct the experiments and then the consequent results.

## 4 SIMULATION SETTING AND RESULTS

### 4.1 SIMPLE REPRESENTATIVE SIMULATION RESULTS

For the purpose of illustration, we first report only a part of our simulation setting pertaining to only one dataset, a part that is representative of the general theme of our results in this paper. This suffices for initiating the motivation required for further theoretical and empirical studies. In the next subsections, we will provide a comprehensive account of our experiments.

In this simulation, we trained 10 autoencoders, one for each class of CIFAR-10 dataset. Our autoencoders incorporate the noise $\bar{z}$ which is a random variable following the standard Gaussian distribution (zero mean and unit variance) that is multiplied by the scalar $\alpha$ for each neuron of each layer of the model as presented in Eq. 1. We use convolutional neural networks (CNNs) with channel numbers of $[30, 73, 100]$ for the encoder, and the mirror of it for the decoder. The stride is 2, padding is 1, and the window size is $4\times4$. We used MSE as the loss function although binary cross-entropy (BCE) has also been tried and produced similar results. The number of samples $n$ is set to 1,000. This simulation setting is of interest in incremental learning where only one or a few classes are present at a time (van de Ven et al., 2021). Moreover, doing classification via generative modeling fits in the framework of predictive coding which is a model of visual processing which proposes that the brain possesses a generative model of input signal with prediction loss contributing as a mean for both learning and attention (Keller & Welling, 2021).

Fig. 3 shows the performances (classification accuracies) in terms of the scalar $\alpha$ of Gaussian noise during both training and test. The omnipresent activation noise which is present in both training and inference exhibits two surprising behaviors in our simulations: (i) it significantly improves the performance (about 200%), which is unprecedented among similar studies in the literature. (ii) The other interesting observation is that always being on or close to the diagonal leads to the best results. These observations will be discussed.

### 4.2 THEORIES FOR THE SIMULATION RESULTS

We first ask the question that why is noise so effective in improving the performance? For that, we need to reiterate that noise is playing two different roles at (i) training and (ii) inference. For the first role, as alluded to before, what actually activation noise is doing during training is regularization. We suggested that indeed activation noise is a general case of dropout. We later demonstrated that activation noise is a form of loss regularization, and also an instance of data augmentation. All these categories fall under the umbrella of regularization. Meanwhile, we learned that, during the test phase, activation noise does sampling which reduces the inference error. In aggregation, therefore, regularization hand in hand with sampling account for this performance boost (about 200%).

The other interesting observation in our simulations is that to attain the highest performance, it is best for the noise scalar $\alpha$ (and in general for the distributions) of training and test noise to be equal. This can be clearly observed in Fig. 3 (and also in all other numerical results that we will present later). We ask the question that why is there such a strong correlation? Shouldn't the two roles of noise during training and test be mutually independent? Where and how do these two roles link to each other? Upon empirical and theoretical investigations, we concluded that the inter-relation between noise during training and test can be characterized and proved in the following theorem.

**Theorem 1** (Noise Distribution Must be the Same During Training and Test)**:** *Via having the inference noise (i.e., the sampler) $r(\boldsymbol{z})$ following the same distribution of noise during training $q^*(\boldsymbol{z})$, i.e., $r^*(\boldsymbol{z}) = q^*(\boldsymbol{z})$, the loss of the inference is minimized.*

*Proof:* During the inference, to optimize the performance of the EBM model, the objective is to find the distribution $r^*(\boldsymbol{z})$ for the activation noise that minimizes the loss specified as follows:

$$r^*(\boldsymbol{z}) = \arg\min_{r(\boldsymbol{z})} \mathcal{L}_{\mathrm{ML}}(\boldsymbol{\theta}^*, r(\boldsymbol{z})), \quad \mathcal{L}_{\mathrm{ML}}(\boldsymbol{\theta}^*, r(\boldsymbol{z})) = \mathbb{E}_{(\boldsymbol{x},y)\sim p_D;\, \boldsymbol{z}\sim r(\boldsymbol{z})}\left[-\log p_{\boldsymbol{\theta}^*}(\boldsymbol{x} \mid y, \boldsymbol{z})\right].$$
(10)

Meanwhile, from Eq. 4 we know that training is performed to minimize the loss $\mathcal{L}_{\mathrm{ML}}(\boldsymbol{\theta}^*, q^*(\boldsymbol{z}))$ with distribution $q^*(\boldsymbol{z})$ as the activation noise. Therefore, during inference, if we set the activation noise $r^*(\boldsymbol{z})$ the same as $q^*(\boldsymbol{z})$ in Eq. 10, the loss again will be minimal. □

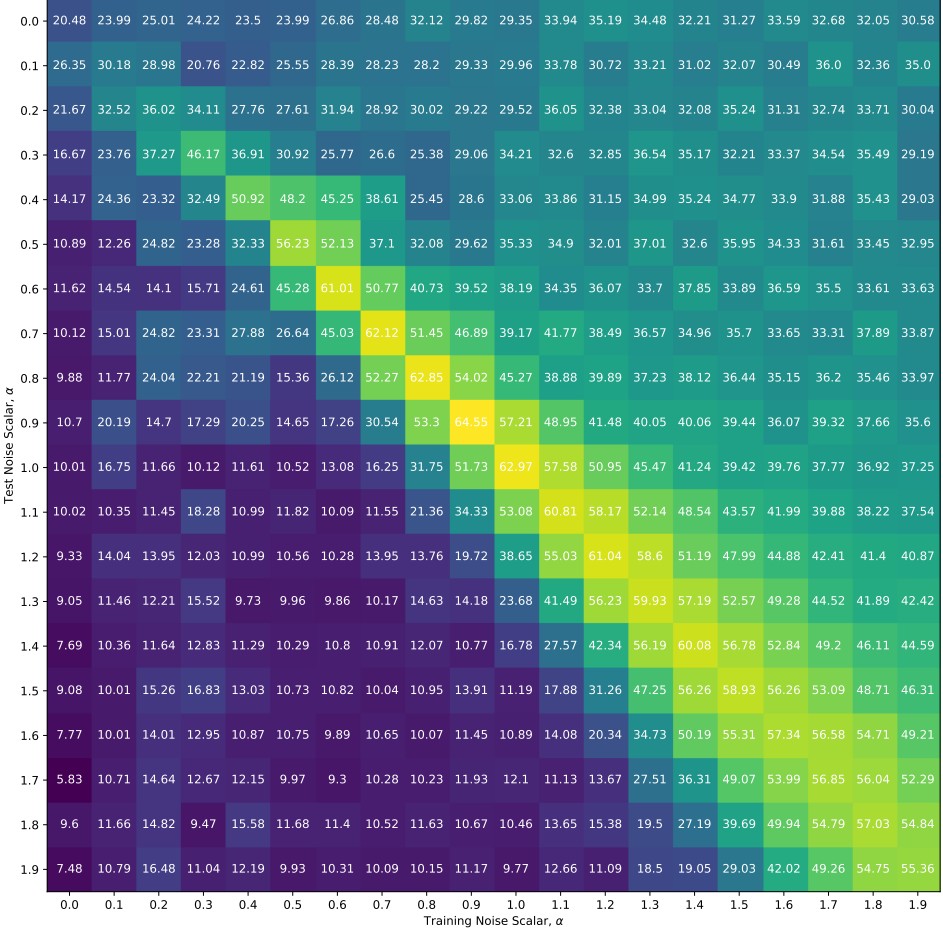

Figure 3: Different values for the scalar $\alpha$ of activation noise during training and test. When the activation noise during training and test are zero (no noise), the classification accuracy is $20.48\%$ whereas at noise coordinate $(0.9, 0.9)$ it is $64.55\%$. This is more than three times increase or $200\%$ improvement in performance. The first and second elements of the noise coordinate represent the values of noise scalar, $\alpha$, in the training and inference, respectively.

This implies that, during training, the distribution of our trained model is impacted by the distribution of noise $q(z)$ via the second term of loss function in Eqs. 7 where the network function learns the distribution of signal in presence of a certain noise distribution. During inference, when the noise distribution is the same as in the training, the performance is the best, and as soon as there is a discrepancy between the two distributions, the performance degrades.

### 4.3 COMPREHENSIVE SIMULATION RESULTS FOR GAUSSIAN NOISE

We present extensive simulation results for various datasets (using Google Colab Pro+) considering Gaussian activation noise. We used the same architecture as the one presented in Section 4.1 for all of our five datasets: CIFAR-10, CIFAR-100 (Krizhevsky et al., 2009), CalTech-101 (Fei-Fei et al., 2004), Flower-102 (Nilsback & Zisserman, 2008), and CUB-200 (Wah et al., 2011). Given that the volume of our experiments is large, for computational tractability we slimmed down the larger datasets from about 100 (or 200) classes to 20 classes.

Our comprehensive simulation results explore the performances of our proposed EBM framework across five dimensions: (i) for five datasets we examined the joint impacts of various noise distributions. The noise scalar $\alpha$ varies during both (ii) training and (iii) test time within the range of $[0, 2)$ for 20 values with the step size of $0.1$. However, we do not report all of them due to the page limit;

Table 1: Accuracies for various datasets, noise scalars $\alpha$, and number of samples $n$. We observe that the performance on all datasets (except for CIFAR-20) peak at the noise scalars $(0.9, 0.9)$. Meanwhile, the number of samples $n$ is set to $10^4$ except for $\alpha = 0$ during inference.

| Scalars, $\alpha$ | Samples | CIFAR-10 | CIFAR-20 | CalTech-20 | Flower-20 | CUB-20 |
|---|---|---|---|---|---|---|
| $(0.0, 0.0)$ | $n = 1$ | $20.48_{\pm 2.03}$ | $14.11_{\pm 2.15}$ | $16.47_{\pm 2.13}$ | $19.61_{\pm 2.21}$ | $18.53_{\pm 2.19}$ |
| $(0.0, 0.5)$ | $n = 10^4$ | $11.63_{\pm 1.74}$ | $18.23_{\pm 2.13}$ | $24.93_{\pm 1.55}$ | $27.01_{\pm 2.03}$ | $21.94_{\pm 1.65}$ |
| $(0.5, 0.0)$ | $n = 1$ | $23.99_{\pm 1.34}$ | $15.91_{\pm 2.33}$ | $17.03_{\pm 2.01}$ | $20.08_{\pm 2.01}$ | $19.39_{\pm 1.71}$ |
| $(0.5, 0.5)$ | $n = 10^4$ | $58.72_{\pm 1.95}$ | $21.44_{\pm 1.78}$ | $25.32_{\pm 1.48}$ | $26.05_{\pm 1.53}$ | $23.85_{\pm 1.41}$ |
| $(0.5, 0.9)$ | $n = 10^4$ | $15.71_{\pm 1.92}$ | $21.97_{\pm 1.69}$ | $20.31_{\pm 1.69}$ | $23.01_{\pm 1.21}$ | $22.03_{\pm 1.37}$ |
| $(0.5, 1.5)$ | $n = 10^4$ | $11.38_{\pm 1.76}$ | $9.64_{\pm 0.84}$ | $10.75_{\pm 0.96}$ | $20.12_{\pm 1.55}$ | $19.35_{\pm 1.15}$ |
| $(0.9, 0.5)$ | $n = 10^4$ | $31.38_{\pm 1.87}$ | $18.98_{\pm 1.73}$ | $19.13_{\pm 1.02}$ | $25.98_{\pm 1.34}$ | $24.61_{\pm 1.41}$ |
| $(0.9, 0.9)$ | $n = 10^4$ | $\mathbf{65.88}_{\pm 1.10}$ | $26.43_{\pm 2.08}$ | $\mathbf{30.32}_{\pm 1.69}$ | $\mathbf{31.05}_{\pm 1.29}$ | $\mathbf{29.83}_{\pm 1.37}$ |
| $(0.9, 1.5)$ | $n = 10^4$ | $14.31_{\pm 2.01}$ | $19.87_{\pm 1.58}$ | $21.32_{\pm 1.67}$ | $23.05_{\pm 1.29}$ | $22.43_{\pm 1.36}$ |
| $(1.5, 0.5)$ | $n = 10^4$ | $36.32_{\pm 1.84}$ | $19.63_{\pm 1.61}$ | $15.25_{\pm 0.83}$ | $23.71_{\pm 1.54}$ | $22.46_{\pm 1.39}$ |
| $(1.5, 0.9)$ | $n = 10^4$ | $40.14_{\pm 1.79}$ | $22.33_{\pm 1.53}$ | $23.43_{\pm 1.43}$ | $22.33_{\pm 1.53}$ | $21.49_{\pm 1.38}$ |
| $(1.5, 1.5)$ | $n = 10^4$ | $59.23_{\pm 2.41}$ | $\mathbf{28.09}_{\pm 1.75}$ | $27.68_{\pm 1.44}$ | $28.42_{\pm 1.65}$ | $24.38_{\pm 1.65}$ |

instead, we selectively present values that are representative of the two key observations which we intend to highlight: these values are $(0.5, 0.5)$, $(0.5, 0.9)$, $(0.5, 1.5)$, $(0.9, 0.5)$, $(0.9, 0.9)$, $(0.9, 1.5)$, and $(1.5, 1.5)$, where the pair $(\cdot, \cdot)$ denotes the scalar $\alpha$ of noise at training and test. (iv) The number of samples $n$ is set to $10^4$, except for the cases of no/zero test noise in which $n = 1$, because when the scalar $\alpha$ of noise during test is 0 (no noise) such as in rows $(0.0, 0.0)$ and $(0.5, 0.0)$, the corresponding accuracy would not vary. Finally, (v) we assess the performances for different noise distributions via exploring all of their permutations.

In our simulations we resize the resolution of all images for our datasets to $32 \times 32$. The number of epochs is set to 100. For optimizer, Adam is chosen with the default learning rate of 0.001 and default momentum (Kingma & Ba, 2014). The minibatch size is 256. We ran 10 experiments for each reported number in Table 1 and we present the means as well as standard error of the means (SEMs) over these runs.

The first observation we intend to emphasize is that this very activation noise can contribute to a considerable improvement in performance as high as $200\%$. This can be discerned in Table 1 when comparing the row $(0.0, 0.0)$ pertaining to the case with no noise, with the row $(0.9, 0.9)$. When going from the former case to the latter case, the accuracy jumps from $20.48\%$ to $65.88\%$ for CIFAR-10 dataset with $n = 10,000$ samples.

The same phenomenon can also be observed for all other datasets. Compared with the previous studies in the literature on the effectiveness of noise, this level of performance enhancement that is acquired by injecting a simple noise is unprecedented if not unimaginable. This performance enhancement, as we theorized and proved in Theorem 1, is a result of the combination of both regularization and sampling, not in an independent way, but indeed in a complicated interdependent fashion (see Fig. 3 to discern the interdependency).

The second observation signifies the importance of having a *balance* in injecting noise at training and test: the noise distribution at test $r(\boldsymbol{z})$ ought to follow the distribution of the noise during training $q(\boldsymbol{z})$ so that the loss of the inference is minimized as discussed in Theorem 1. This can be noticed when contrasting *balanced* rows $(0.5, 0.5)$, $(0.9, 0.9)$, and $(1.5, 1.5)$ with the rest which are unbalanced. Interestingly, even the row $(0.0, 0.0)$ which enjoys no noise outperforms unbalanced rows $(0.0, 0.5)$, $(0.5, 0.9)$, $(0.5, 1.5)$, and $(0.9, 1.5)$.

The third observation is the asymmetry between having unbalanced noise for training versus test. Too large a scalar $\alpha$ for noise during test has far more negative impact on the performance than otherwise. For example, consider rows $(0.5, 0.0)$ and $(0.0, 0.5)$, the former (large noise scalar $\alpha$ at training) delivers about $100\%$ higher accuracies than the latter which bears high noise scalar $\alpha$ at test. This pattern, with no exception, repeats in all rows where the noise scalar during test is larger than that of the training noise such as $(0.0, 0.5)$, $(0.5, 0.9)$, $(0.5, 1.5)$, and $(0.9, 1.5)$. All these cases yield poorest performances even worse than $(0.0, 0.0)$. This observation (as well as the preceding two observations) are better discerned in Fig. 3 which portrays the heatmap of performance results.

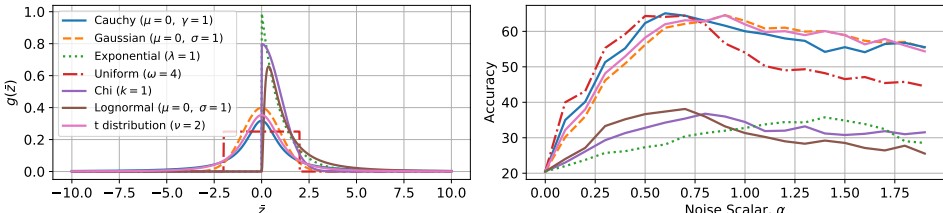

Figure 4: On the left side, we see different profiles pertaining to various noise distributions. On the right side, the accuracies for different noise distributions are shown as the scalar $\alpha$ increases.

## 4.4 INVESTIGATING OTHER NOISE DISTRIBUTIONS

We now evaluate the performances of alternative noise distributions. We consider both types of distributions, symmetric and asymmetric: (i) Cauchy, (ii) uniform, (iii) Chi, (iv) lognormal, (v) t-Distribution, and (vi) exponential. For noise distributions that are capable of becoming either symmetric or not (e.g., uniform and Cauchy), we explore parameters that keep them symmetric around zero; because based on our experiments (as we will see) we are convinced that symmetric distributions (i.e., symmetric about zero) consistently yield higher performances.

For different noise distributions, the underlying rule that justifies the results can be summarized as follows: when the activation noise $z$ is added to the signal $s$, it is conducive to the performance as far as it does not entirely distort the signal. In other words, the Signal-to-Noise Ratio (SNR) can be beneficially decreased to a threshold that delivers the peak performance; after that the performance begins to degrade. Given that $t = s + z = s + \alpha\bar{z}$, and $s$ is constant, there are two factors that determine the value of the SNR (given by $\mathrm{var}[s]/\mathrm{var}[z]$): (i) the distribution of $\bar{z}$, particularly how narrowly (or broadly) the density of the random variable is distributed (i.e., $\mathrm{var}[\bar{z}]$); (ii) how large the scalar of $\alpha$ is. Fig. 4 presents the profile of different probability distributions: note the breadth of various probability distributions as they play a vital role in how well they perform. In the following we discuss our results with respect to the value of the SNR.

As we can see in Fig. 4, (i) adopting symmetric probability distributions as activation noise substantially outperforms asymmetric ones: hence, we observe that Cauchy, Gaussian, uniform, and t distribution consistently yield higher accuracies than Chi, exponential, and lognormal. This is because having a symmetric sampling can more effectively explore the learned manifold of the neural network. (ii) The performances of the different noise distributions can be justified with respect to the value of the SNR as follows: as we reduce the SNR via increasing the noise scalar $\alpha$, the performance rises up to a peak and then falls; this is valid for all the probability distributions. The differences in the slope of various distributions (in Fig. 4 on right) originate from the difference in the profile of the noise distributions: the wider the noise profile is (the larger $\mathrm{var}[\bar{z}]$), the sooner the SNR will drop, and the earlier the pattern of rise and fall will occur. For example, in Fig. 4 because the Gaussian distribution is thinner than the Cauchy, its pattern of rise and fall happens with a small delay since its SNR drops with a latency.

## 5 CONCLUSION

In this paper, specifically for an EBM setting, we studied the impacts of activation noise during training time, broached its role at the inference time, and scrutinized the relationship between activation noise at training and inference. We proved that, during training, activation noise is a general form of dropout; we discussed the relationship between activation noise and loss regularization/data augmentation. We studied the empirical/theoretical aspects of activation noise at inference time and proved that adopting noisy activation functions, during inference, can be interpreted as sampling the model. We proved/demonstrated that when the sampler follows the same distribution as the noise during training, the loss of the inference is minimized and therefore the performance is optimized. To assess our theoretical results, we performed extensive experiments exploring different datasets, noise scalars during training and inference, and different noise distributions.

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

# Appendix

With all that has been said on activation noise, however, there still remains a number of results unreported and many critical questions left unanswered due to the space limit: in this supplementary material, (a) we report some leftover results elaborating on those results given in Section 4.3. (b) We answer the question that what is the impact of the number of samples $n$ on the performance? (c) In the main text, we used the same noise distribution during both training and inference but what if the noise distributions differ during training and inference? (d) We applied activation noise to all layers of the network but what if only a portion of layers are injected with noise? How about exploring the case where both the noise scalar $\alpha$ and distribution vary at different layers? (e) We studied the effectiveness of activation noise specifically for a generative EBM setting; but, what about the typical discriminative modeling image classification scenario? Is activation noise also effective for that? We will answer all these questions comprehensively in this supplementary material exactly in the alphabetical order that they were presented above: we perform extensive experiments exploring many simulation scenarios pertaining to questions asked above.

## A    MORE DETAILS FOR THE PERFORMANCE RESULTS OF VARIOUS NOISE DISTRIBUTIONS

In the following we provide the specifications of our probability distributions: as shown in Fig. 5 for Cauchy distribution $g(\bar{z}; z_0, \gamma)$, we set $z_0$, which characterizes the mean of the Cauchy distribution, equal to zero; and explore different values of $\gamma$. The same way for uniform distribution, the mean is set to zero: while uniform distribution originally has two parameters characterizing the start and end of the random variable, in our simulation, because the uniform noise is desired to be symmetric, we define only one parameter characterizing the uniform distribution denoted by $\omega$ referring to its width. For Chi distribution $g(\bar{z}; k)$ we assess its performance with different values of $k$, whereas for lognormal distribution $g(\bar{z}; \mu, \sigma)$, the value of $\mu$ is set to zero and $\sigma$ is varied. For t-distribution $g(\bar{z}; \nu)$, different values of $\nu$ are explored. Worth mentioning that for t-distribution when $\nu = 1$, the distribution reduces to Cauchy distribution. Finally, for exponential distribution $g(\bar{z}; \lambda)$, we evaluate the performance for different values of $\lambda$. Our experiments are performed 10 times via multiple random seeds. We report the means ($\pm$ SEMs) over these experiments.

Overall, based on the performances across all values of the noise scalar $\alpha$ (as shown in Fig. 6), we conclude that the best noise distribution is also the most popular one: Gaussian distribution; perhaps because it has the largest area under the curve for different values of the noise scalar $\alpha$. Furthermore, it is clear that the distributions that are similar to the Gaussian distribution deliver best performances whereas for dissimilar ones the performance becomes worse. Meanwhile, we realized that various parameters (standard deviations $\sigma$) for the Gaussian distribution do not offer convincing improvements and only hasten/delay the occurrence of the rise and fall pattern for classification accuracy.

## B    IMPACT OF THE NUMBER OF SAMPLES

In Fig. 7 we demonstrate the impact of the number of samples for the standard Gaussian noise $\bar{z}$ with different noise scalar values $\alpha$ during training and test. It can be seen that the accuracy rises almost logarithmically as the number of samples increases. Note that these results pertain to CIFAR-10 dataset. In Table 2 we provide the comprehensive form of the Table 1 including the classification accuracies for different number of samples $n$.

## C    DIFFERENT NOISE DISTRIBUTIONS DURING TRAINING AND INFERENCE

We examine the performances pertaining to the combination of nonidentical noise distributions: adopting two different distributions, one for training and the other for inference. The parameters of different noise distributions are outlined in Table 3. The noise scalar $\alpha$ is set to one for training and test. The results of the experiment are displayed in Fig. 8 that demonstrates the performances for each two noise distributions in a $7\times7$ heatmap grid: the empirical results confirm our theoretical

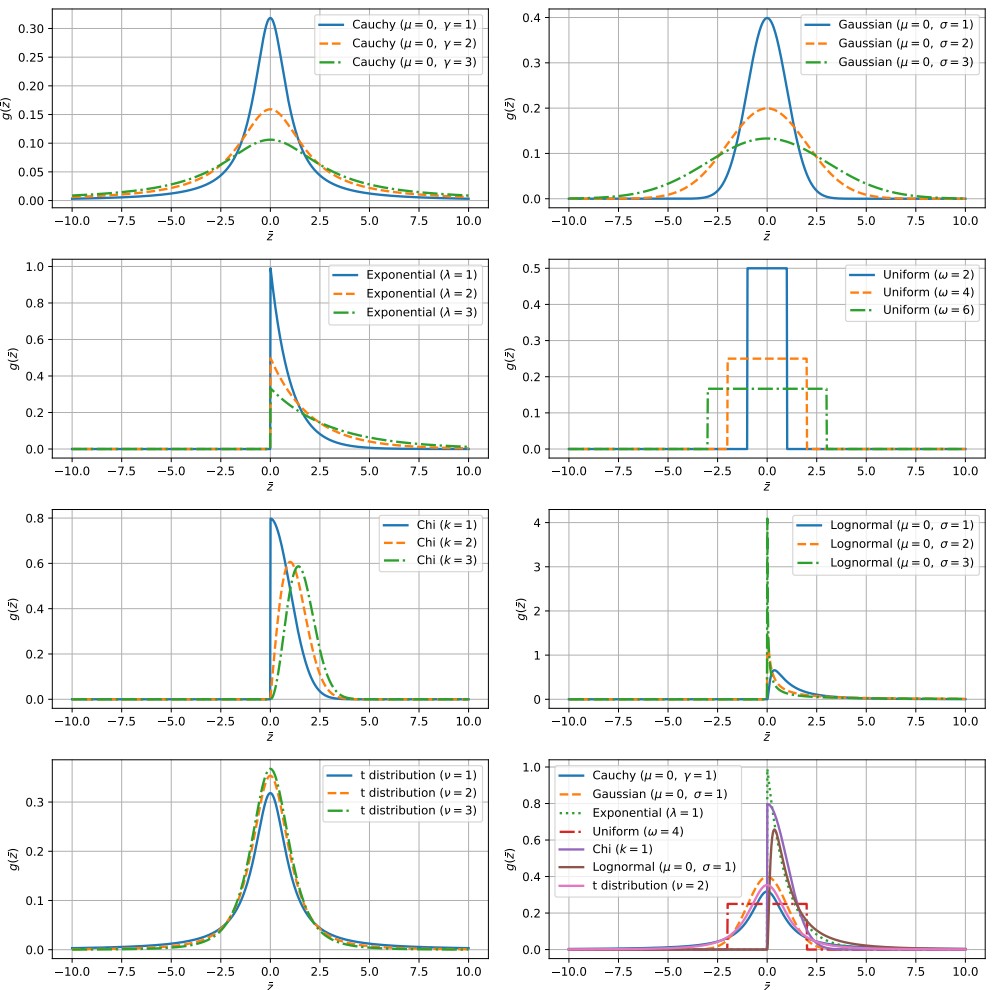

Figure 5: This figure shows and juxtaposes different probability distributions that were chosen for activation noise. Barring the uniform distribution, all other distributions have heavier tails than that of Gaussian distribution. We believe having a heavy tail hastens the pattern of rise and fall of the classification accuracy as it increases the SNR value. Meanwhile, symmetric distributions with high margin outperform asymmetric ones.

conclusions that was proposed in Theorem 1, proving that the optimal performance during inference is acquired if we have the same noise distributions during training and inference.

In Fig. 8 it can be seen that the best results pertain to our four symmetric noise distributions, whereas the worst are for when an asymmetric noise is adopted for training and a symmetric noise for inference (to sample). This result is in accordance with our third observation discussed in Section 4.3 and also further corroborates Theorem 1. In Fig. 8, we can see that if we use symmetric noise during training and asymmetric noise during inference the performance will not be as poor as otherwise. This was also observed and explained in Section 4.4.

## D    LAYER-SELECTIVE NOISE FOR THE EBM SETTING

So far, we have injected noise to all the layers of our neural networks. In this section, we study the case where we inject noise only to a portion of layers of the generative EBM model. We design and compare five schemes for noise injection: (i) full noise (the default) where noise is injected to all layers (denoted by n-full); (ii) and (iii) odd/even noise where noise is injected to all odd/even

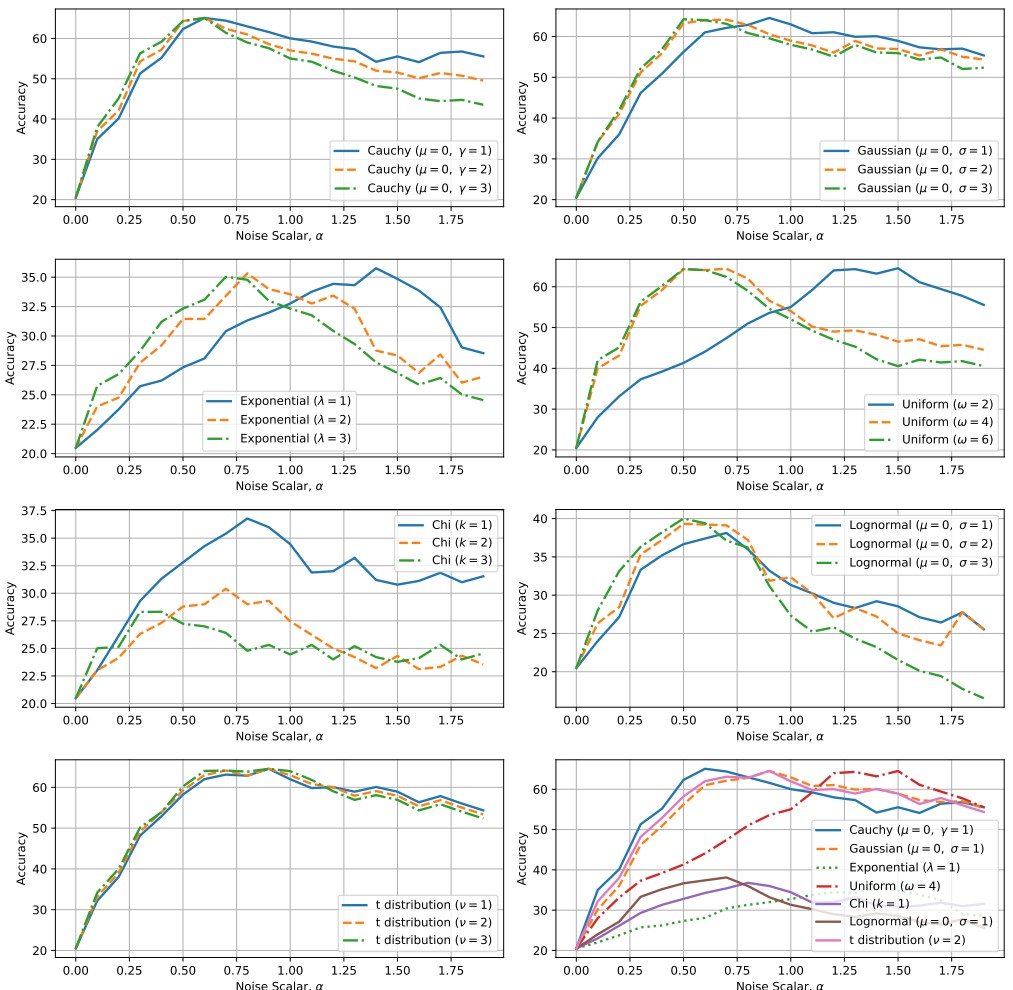

Figure 6: In this figure we demonstrate the performances of different distributions both symmetric and asymmetric for three different cases of parameters in each individual figure. Then at the bottom-right figure we compare the best performances pertaining to the best parameters of each individual figure.

layers (denoted by n-odd and n-even); (iv) and (v) encoder/decoder noise where noise is injected to encoder/decoder layers (denoted by n-enc and n-dec). Worth mentioning that we used standard Gaussian noise with scalar $\alpha = 1$ pertaining to the peak performance for both training and test.

Fig. 9 presents the accuracies for our neural network injected with all the noise schemes presented above. As it can be seen, injecting noise to all layers yields higher performances than the alternatives; n-even and n-odd are at the next rank while n-enc and n-dec are the worse. Fig. 9 also jointly investigates the impact of injecting noise to different layers of neural network at both training and inference. We observe the results akin to the previous results in Fig. 3 once again *but for each layer*: according to our results the noise injected to a *specific layer* during training and inference better to follow the same distribution. For example, when noise is injected to even layers (n-even) of the neural network during training, none of the other noise injection schemes during inference could exceed the performance of n-even noise injection. Also, not injecting noise during training for a specific layer while injecting noise during inference for that layer significantly reduces the performance. Accordingly, inspired by above results we can propose Corollary 1 that generalizes Theorem 1.

Table 2: Accuracies for various datasets, different values of scalar noise $\alpha$, and a range of value for the number of samples $n$. The pair $(\cdot, \cdot)$ in the left-most column denotes the scalar $\alpha$ of noise at training and test.

| Scalars, $\alpha$ | Samples | CIFAR-10 | CIFAR-20 | CalTech-20 | Flower-20 | CUB-20 |
|---|---|---|---|---|---|---|
| $(0.0, 0.0)$ | $n = 1$ | $20.48_{\pm 2.03}$ | $14.11_{\pm 2.15}$ | $16.47_{\pm 2.13}$ | $19.61_{\pm 2.21}$ | $18.53_{\pm 2.19}$ |
| | $n = 1$ | $9.12_{\pm 2.25}$ | $13.63_{\pm 2.49}$ | $15.43_{\pm 2.18}$ | $19.45_{\pm 2.25}$ | $18.32_{\pm 2.27}$ |
| | $n = 10$ | $9.76_{\pm 2.22}$ | $15.95_{\pm 2.47}$ | $18.57_{\pm 2.05}$ | $21.71_{\pm 2.23}$ | $19.13_{\pm 2.19}$ |
| $(0.0, 0.5)$ | $n = 10^2$ | $9.98_{\pm 1.92}$ | $16.43_{\pm 2.42}$ | $20.03_{\pm 1.91}$ | $23.81_{\pm 2.14}$ | $19.81_{\pm 1.85}$ |
| | $n = 10^3$ | $10.89_{\pm 1.85}$ | $17.05_{\pm 2.37}$ | $22.59_{\pm 1.75}$ | $25.32_{\pm 2.07}$ | $20.32_{\pm 1.83}$ |
| | $n = 10^4$ | $11.63_{\pm 1.74}$ | $18.23_{\pm 2.13}$ | $24.93_{\pm 1.55}$ | $27.01_{\pm 2.03}$ | $21.94_{\pm 1.65}$ |
| $(0.5, 0.0)$ | $n = 1$ | $23.99_{\pm 1.34}$ | $15.91_{\pm 2.33}$ | $17.03_{\pm 2.01}$ | $20.08_{\pm 2.01}$ | $19.39_{\pm 1.71}$ |
| | $n = 1$ | $23.25_{\pm 2.32}$ | $15.93_{\pm 2.44}$ | $16.09_{\pm 1.93}$ | $19.77_{\pm 2.07}$ | $18.36_{\pm 2.13}$ |
| | $n = 10$ | $36.04_{\pm 2.19}$ | $17.63_{\pm 2.23}$ | $19.01_{\pm 1.84}$ | $21.04_{\pm 1.94}$ | $19.91_{\pm 2.09}$ |
| $(0.5, 0.5)$ | $n = 10^2$ | $42.75_{\pm 2.14}$ | $19.84_{\pm 2.11}$ | $21.37_{\pm 1.74}$ | $23.07_{\pm 1.74}$ | $21.26_{\pm 1.93}$ |
| | $n = 10^3$ | $56.23_{\pm 2.25}$ | $20.93_{\pm 1.93}$ | $23.02_{\pm 1.59}$ | $24.85_{\pm 1.64}$ | $22.49_{\pm 1.72}$ |
| | $n = 10^4$ | $58.72_{\pm 1.95}$ | $21.44_{\pm 1.78}$ | $25.32_{\pm 1.48}$ | $26.05_{\pm 1.53}$ | $23.85_{\pm 1.41}$ |
| | $n = 1$ | $12.25_{\pm 2.24}$ | $17.77_{\pm 2.27}$ | $14.45_{\pm 2.06}$ | $19.08_{\pm 2.10}$ | $18.43_{\pm 2.23}$ |
| | $n = 10$ | $12.94_{\pm 2.22}$ | $18.89_{\pm 2.23}$ | $16.91_{\pm 2.14}$ | $20.94_{\pm 1.97}$ | $19.49_{\pm 2.08}$ |
| $(0.5, 0.9)$ | $n = 10^2$ | $13.82_{\pm 2.14}$ | $20.45_{\pm 2.19}$ | $18.03_{\pm 2.03}$ | $21.51_{\pm 1.64}$ | $20.51_{\pm 1.94}$ |
| | $n = 10^3$ | $14.65_{\pm 2.01}$ | $21.34_{\pm 2.08}$ | $19.42_{\pm 1.79}$ | $22.95_{\pm 1.34}$ | $21.38_{\pm 1.74}$ |
| | $n = 10^4$ | $15.71_{\pm 1.92}$ | $21.97_{\pm 1.69}$ | $20.31_{\pm 1.69}$ | $23.01_{\pm 1.21}$ | $22.03_{\pm 1.37}$ |
| | $n = 1$ | $9.37_{\pm 2.43}$ | $7.66_{\pm 1.05}$ | $7.93_{\pm 1.12}$ | $17.61_{\pm 2.21}$ | $16.53_{\pm 2.03}$ |
| | $n = 10$ | $9.79_{\pm 2.34}$ | $8.03_{\pm 1.03}$ | $8.52_{\pm 1.08}$ | $18.13_{\pm 2.13}$ | $17.07_{\pm 1.95}$ |
| $(0.5, 1.5)$ | $n = 10^2$ | $10.01_{\pm 2.13}$ | $8.89_{\pm 1.01}$ | $9.04_{\pm 1.02}$ | $18.98_{\pm 1.94}$ | $18.39_{\pm 1.88}$ |
| | $n = 10^3$ | $10.73_{\pm 2.04}$ | $9.36_{\pm 0.89}$ | $9.93_{\pm 0.98}$ | $19.53_{\pm 1.63}$ | $18.71_{\pm 1.32}$ |
| | $n = 10^4$ | $11.38_{\pm 1.76}$ | $9.64_{\pm 0.84}$ | $10.75_{\pm 0.96}$ | $20.12_{\pm 1.55}$ | $19.35_{\pm 1.15}$ |
| | $n = 1$ | $25.98_{\pm 2.34}$ | $16.19_{\pm 2.24}$ | $11.22_{\pm 1.51}$ | $19.91_{\pm 2.11}$ | $18.78_{\pm 2.00}$ |
| | $n = 10$ | $27.50_{\pm 2.17}$ | $16.47_{\pm 2.13}$ | $13.87_{\pm 1.36}$ | $21.40_{\pm 1.99}$ | $20.00_{\pm 1.87}$ |
| $(0.9, 0.5)$ | $n = 10^2$ | $28.81_{\pm 2.11}$ | $17.33_{\pm 2.11}$ | $15.35_{\pm 1.20}$ | $23.33_{\pm 1.71}$ | $21.56_{\pm 1.48}$ |
| | $n = 10^3$ | $29.62_{\pm 2.09}$ | $18.05_{\pm 2.09}$ | $17.61_{\pm 1.06}$ | $24.76_{\pm 1.56}$ | $23.28_{\pm 1.67}$ |
| | $n = 10^4$ | $31.38_{\pm 1.87}$ | $18.98_{\pm 1.73}$ | $19.13_{\pm 1.02}$ | $25.98_{\pm 1.34}$ | $24.61_{\pm 1.41}$ |
| | $n = 1$ | $25.25_{\pm 2.40}$ | $22.54_{\pm 0.24}$ | $24.03_{\pm 2.11}$ | $22.52_{\pm 2.10}$ | $21.13_{\pm 1.99}$ |
| | $n = 10$ | $42.04_{\pm 2.37}$ | $23.21_{\pm 2.23}$ | $26.41_{\pm 2.03}$ | $24.94_{\pm 2.04}$ | $23.45_{\pm 1.78}$ |
| $(0.9, 0.9)$ | $n = 10^2$ | $53.02_{\pm 1.34}$ | $24.37_{\pm 2.19}$ | $28.44_{\pm 1.99}$ | $26.58_{\pm 1.89}$ | $25.62_{\pm 1.56}$ |
| | $n = 10^3$ | $64.55_{\pm 1.15}$ | $25.25_{\pm 2.11}$ | $29.32_{\pm 1.87}$ | $29.45_{\pm 1.35}$ | $28.45_{\pm 1.43}$ |
| | $n = 10^4$ | $\mathbf{65.88}_{\pm 1.10}$ | $26.43_{\pm 2.08}$ | $\mathbf{30.32}_{\pm 1.69}$ | $\mathbf{31.05}_{\pm 1.29}$ | $\mathbf{29.83}_{\pm 1.37}$ |
| | $n = 1$ | $11.37_{\pm 2.40}$ | $17.41_{\pm 2.14}$ | $16.34_{\pm 2.21}$ | $19.88_{\pm 2.10}$ | $19.03_{\pm 2.04}$ |
| | $n = 10$ | $12.24_{\pm 2.37}$ | $17.41_{\pm 2.11}$ | $17.54_{\pm 2.14}$ | $20.24_{\pm 2.04}$ | $19.59_{\pm 1.96}$ |
| $(0.9, 1.5)$ | $n = 10^2$ | $13.01_{\pm 2.24}$ | $18.13_{\pm 2.01}$ | $18.66_{\pm 2.02}$ | $21.01_{\pm 1.84}$ | $20.72_{\pm 1.75}$ |
| | $n = 10^3$ | $13.91_{\pm 2.15}$ | $19.01_{\pm 1.88}$ | $20.07_{\pm 1.79}$ | $22.35_{\pm 1.59}$ | $21.61_{\pm 1.47}$ |
| | $n = 10^4$ | $14.31_{\pm 2.01}$ | $19.87_{\pm 1.58}$ | $21.32_{\pm 1.67}$ | $23.05_{\pm 1.29}$ | $22.43_{\pm 1.36}$ |
| | $n = 1$ | $18.75_{\pm 2.21}$ | $15.73_{\pm 2.15}$ | $9.35_{\pm 1.33}$ | $18.61_{\pm 2.21}$ | $17.38_{\pm 2.18}$ |
| | $n = 10$ | $33.51_{\pm 2.13}$ | $16.47_{\pm 2.11}$ | $11.56_{\pm 1.21}$ | $20.13_{\pm 2.05}$ | $19.28_{\pm 2.02}$ |
| $(1.5, 0.5)$ | $n = 10^2$ | $34.18_{\pm 2.07}$ | $17.35_{\pm 2.01}$ | $13.03_{\pm 1.04}$ | $21.34_{\pm 1.93}$ | $20.52_{\pm 1.86}$ |
| | $n = 10^3$ | $35.95_{\pm 2.04}$ | $18.75_{\pm 1.99}$ | $14.78_{\pm 0.89}$ | $22.13_{\pm 1.74}$ | $21.37_{\pm 1.58}$ |
| | $n = 10^4$ | $36.32_{\pm 1.84}$ | $19.63_{\pm 1.61}$ | $15.25_{\pm 0.83}$ | $23.71_{\pm 1.54}$ | $22.46_{\pm 1.39}$ |
| | $n = 1$ | $31.44_{\pm 0.25}$ | $18.92_{\pm 0.20}$ | $17.07_{\pm 2.08}$ | $18.92_{\pm 2.20}$ | $17.57_{\pm 2.07}$ |
| | $n = 10$ | $36.75_{\pm 0.10}$ | $19.47_{\pm 2.03}$ | $19.03_{\pm 1.95}$ | $19.47_{\pm 2.03}$ | $18.73_{\pm 1.86}$ |
| $(1.5, 0.9)$ | $n = 10^2$ | $37.24_{\pm 0.10}$ | $20.34_{\pm 1.93}$ | $21.43_{\pm 1.78}$ | $20.34_{\pm 1.93}$ | $19.61_{\pm 1.72}$ |
| | $n = 10^3$ | $39.44_{\pm 1.81}$ | $21.15_{\pm 1.85}$ | $22.12_{\pm 1.55}$ | $21.15_{\pm 1.85}$ | $20.33_{\pm 1.63}$ |
| | $n = 10^4$ | $40.14_{\pm 1.79}$ | $22.33_{\pm 1.53}$ | $23.43_{\pm 1.43}$ | $22.33_{\pm 1.53}$ | $21.49_{\pm 1.38}$ |
| | $n = 1$ | $31.53_{\pm 2.29}$ | $24.37_{\pm 2.15}$ | $23.68_{\pm 2.05}$ | $20.28_{\pm 2.14}$ | $19.39_{\pm 2.31}$ |
| | $n = 10$ | $38.72_{\pm 2.19}$ | $25.15_{\pm 1.91}$ | $24.34_{\pm 1.96}$ | $21.31_{\pm 2.01}$ | $20.47_{\pm 2.26}$ |
| $(1.5, 1.5)$ | $n = 10^2$ | $43.98_{\pm 2.55}$ | $26.36_{\pm 1.83}$ | $25.62_{\pm 1.81}$ | $23.27_{\pm 1.97}$ | $22.01_{\pm 2.03}$ |
| | $n = 10^3$ | $58.93_{\pm 2.47}$ | $27.15_{\pm 1.78}$ | $26.37_{\pm 1.69}$ | $25.59_{\pm 1.74}$ | $23.85_{\pm 1.95}$ |
| | $n = 10^4$ | $59.23_{\pm 2.41}$ | $\mathbf{28.09}_{\pm 1.75}$ | $27.68_{\pm 1.44}$ | $28.42_{\pm 1.65}$ | $24.38_{\pm 1.65}$ |

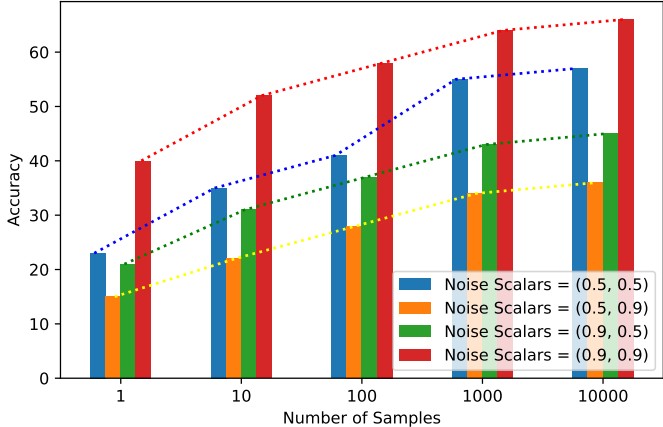

Figure 7: The impact of five different number of samples $n$ on the classification accuracy for different noise coordinates: the pair $(\cdot, \cdot)$ in the legendbox denotes the scalar $\alpha$ of noise at training and test. It appears that the accuracy rises almost logarithmically as the number of samples increases. This can also be seen in Table 2.

Table 3: The parameters used in the experiment pertaining to Fig. 8. We chose the parameters that lead to the best performances of different distributions. The noise scalar $\alpha$ is set to one for training and test.

| | Gaussian | Cauchy | t | uniform | lognormal | Chi | exponential |
|---|---|---|---|---|---|---|---|
| Parameters | $\mu = 0, \sigma = 1$ | $\mu = 0, \gamma = 1$ | $\nu = 1$ | $\omega = 0$ | $\mu = 0, \sigma = 1$ | $k = 1$ | $\lambda = 1$ |

**Corollary 1** (*General* Noise Distribution for the Sampler During Test)**:** *Via having the sampler $r(z, v)$ following both the same placement and distribution of noise during training $q^*(z, v)$, i.e., $r^*(z, v) = q^*(z, v)$, where $v$ encodes the placement of noise among the layers, the loss of the inference is minimized.*

Clearly, when the placement parameter $v$ is selected such that all layers are included for noise injection, Corollary 1 reduces to Theorem 1.

## E   ACTIVATION NOISE IN DISCRIMINATIVE MODELING

So far, we have considered only the generative EBM modeling. In this section, we examine the impact of activation noise on the performance of classifiers relying on discriminative modeling: we conduct our study with two popular datasets, CIFAR-10 and CIFAR-100, on seventeen architectures as listed in Table 4. In this experiment, we train for 30 epochs, the optimizer was Adam with learning rate 0.001 and the default momentum, and the learning rate scheduler is cosine annealing similarity. Fig. 10 demonstrates the results of our experiments: as we can see in Fig. 10 exhibiting the results of the CIFAR-10 dataset for different architectures, the performances reliably deteriorate as we increase the scalar of the omnipresent (standard Gaussian) activation noise *even* with *balanced* noise enjoying $n = 1000$ samples.

Meanwhile, in Table 5, we report the performances of all architectures for both CIFAR-10 and CIFAR-100 with three noise levels: (i) no noise $\alpha = 0.0$, (ii) mild noise $\alpha = 0.5$, and (iii) strong noise $\alpha = 1.0$. Based on our observations, we are convinced that adding even a mild level of balanced noise is detrimental to the performance of the discriminative modeling scenario for different architectures and datasets.

To also investigate the scenario for *unbalanced* noise injection, we provide Fig. 11 that shows the performance heatmap of the discriminative modeling classifier using ResNet50 architecture on CIFAR-10 dataset as we increase the noise scalar $\alpha$ during both the training and inference. We can

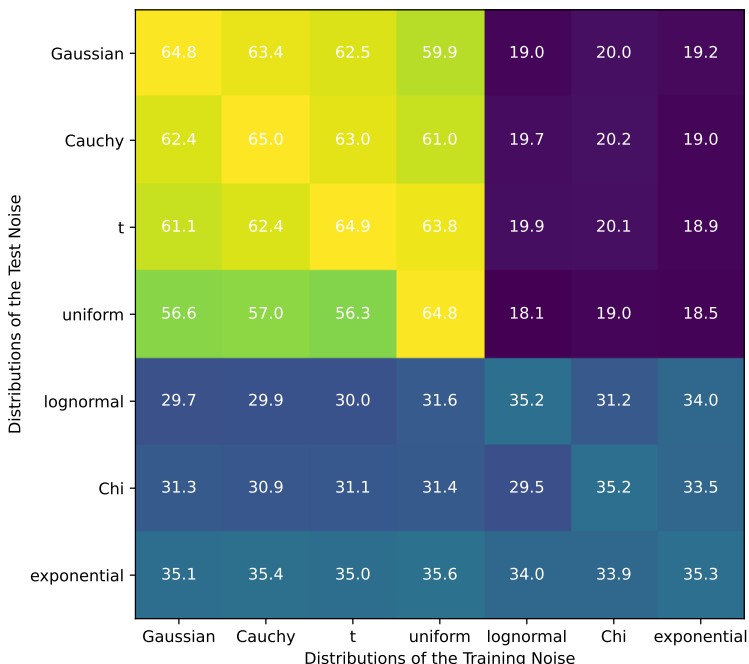

Figure 8: Accuracy of different noise distributions for the unbalanced case. The horizontal axis pertains to the noise distributions during training while the vertical axis corresponds to the noise distributions during inference. We can see that the distribution better be kept the same as the training during inference to guarantee the optimal performance.

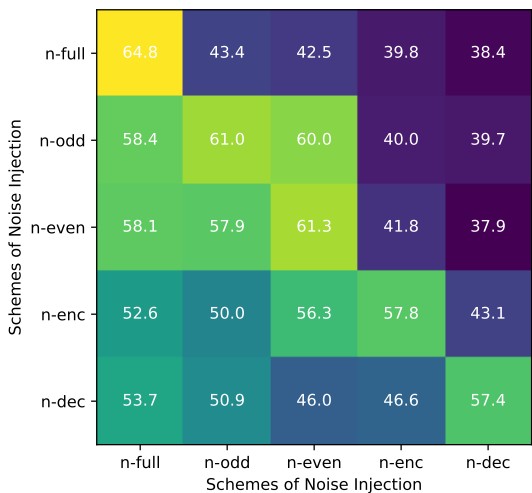

Figure 9: Accuracy of the selective noise injection for the balanced case.

Table 4: Architectures for studying the impact of activation noise on discriminative modeling.

| VGG11 | VGG13 | VGG16 | VGG19 | ResNet18 |
|---|---|---|---|---|
| ResNet50 | ResNet101 | RegNetX-200MF | RegNetX-400MF | MobileNetV2 |
| ResNeXt29(32x4d) | ResNeXt29(2x64d) | SimpleDLA | DenseNet121 | PreActResNet18 |
| DPN92 | DLA | - | - | - |

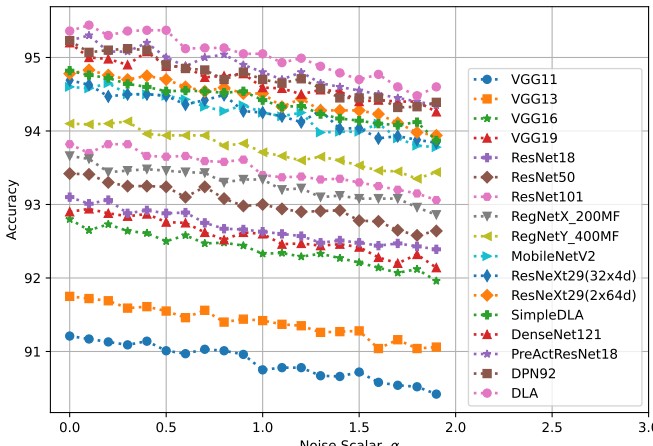

Figure 10: The classification accuracy of different architectures as noise scalar increases. The scalar $\alpha$ for the standard Gaussian noise $\bar{z}$ is set the same during both training and inference.

Table 5: Accuracies for CIFAR-10 and CIFAR-100 datasets, for various balanced noise scalars, and different architectures. The activation noise was injected both during training and inference. The scalar $\alpha$ for the standard Gaussian noise $\bar{z}$ is set the same during both training and inference.

| | Noise Scalars | | | | | |
| | $\alpha = 0.0$ (No Noise) | | $\alpha = 0.5$ (Mild Noise) | | $\alpha = 1.0$ (Strong Noise) | |
| Architectures | CIFAR-10 | CIFAR-100 | CIFAR-10 | CIFAR-100 | CIFAR-10 | CIFAR-100 |
|---|---|---|---|---|---|---|
| VGG11 | $91.21_{\pm0.03}$ | $70.48_{\pm0.03}$ | $91.01_{\pm0.05}$ | $70.16_{\pm0.07}$ | $90.72_{\pm0.06}$ | $69.92_{\pm0.05}$ |
| VGG13 | $91.76_{\pm0.08}$ | $71.02_{\pm0.03}$ | $91.41_{\pm0.04}$ | $70.69_{\pm0.05}$ | $91.23_{\pm0.07}$ | $70.45_{\pm0.05}$ |
| VGG16 | $92.64_{\pm0.03}$ | $71.86_{\pm0.04}$ | $92.39_{\pm0.01}$ | $71.53_{\pm0.02}$ | $92.13_{\pm0.05}$ | $71.23_{\pm0.05}$ |
| VGG19 | $92.91_{\pm0.04}$ | $72.18_{\pm0.04}$ | $92.75_{\pm0.03}$ | $71.89_{\pm0.05}$ | $92.45_{\pm0.07}$ | $70.64_{\pm0.03}$ |
| ResNet18 | $93.02_{\pm0.04}$ | $72.59_{\pm0.03}$ | $92.82_{\pm0.04}$ | $72.27_{\pm0.05}$ | $92.57_{\pm0.03}$ | $71.72_{\pm0.04}$ |
| ResNet50 | $93.62_{\pm0.02}$ | $73.06_{\pm0.01}$ | $93.46_{\pm0.05}$ | $72.78_{\pm0.04}$ | $93.11_{\pm0.05}$ | $72.45_{\pm0.04}$ |
| ResNet101 | $93.75_{\pm0.02}$ | $73.19_{\pm0.02}$ | $93.56_{\pm0.04}$ | $72.91_{\pm0.03}$ | $93.17_{\pm0.04}$ | $72.60_{\pm0.05}$ |
| RegNeXt200MF | $94.24_{\pm0.03}$ | $73.63_{\pm0.03}$ | $94.03_{\pm0.05}$ | $73.34_{\pm0.05}$ | $93.78_{\pm0.06}$ | $73.00_{\pm0.04}$ |
| RegNeXt400MF | $94.29_{\pm0.03}$ | $73.68_{\pm0.02}$ | $94.11_{\pm0.05}$ | $73.39_{\pm0.04}$ | $93.80_{\pm0.07}$ | $73.07_{\pm0.04}$ |
| MobileNetV2 | $94.43_{\pm0.02}$ | $73.83_{\pm0.05}$ | $94.43_{\pm0.02}$ | $73.58_{\pm0.03}$ | $93.93_{\pm0.05}$ | $73.22_{\pm0.03}$ |
| ResNeXt29(32x4d) | $94.73_{\pm0.07}$ | $74.11_{\pm0.06}$ | $94.55_{\pm0.04}$ | $73.83_{\pm0.03}$ | $94.25_{\pm0.03}$ | $73.55_{\pm0.03}$ |
| ResNeXt29(2x64d) | $94.82_{\pm0.04}$ | $74.39_{\pm0.01}$ | $94.63_{\pm0.06}$ | $74.06_{\pm0.05}$ | $94.26_{\pm0.07}$ | $73.75_{\pm0.04}$ |
| SimpleDLA | $94.89_{\pm0.02}$ | $74.43_{\pm0.05}$ | $94.77_{\pm0.05}$ | $74.11_{\pm0.03}$ | $94.30_{\pm0.07}$ | $73.85_{\pm0.06}$ |
| DenseNet121 | $95.04_{\pm0.04}$ | $74.62_{\pm0.04}$ | $94.83_{\pm0.06}$ | $74.29_{\pm0.06}$ | $94.54_{\pm0.07}$ | $74.00_{\pm0.03}$ |
| PreActResNet18 | $95.11_{\pm0.04}$ | $74.88_{\pm0.02}$ | $94.95_{\pm0.06}$ | $74.51_{\pm0.05}$ | $94.63_{\pm0.07}$ | $74.66_{\pm0.04}$ |
| DPN92 | $95.16_{\pm0.03}$ | $74.95_{\pm0.04}$ | $94.99_{\pm0.05}$ | $74.66_{\pm0.07}$ | $94.75_{\pm0.05}$ | $74.33_{\pm0.01}$ |
| DLA | $95.47_{\pm0.05}$ | $75.31_{\pm0.05}$ | $95.23_{\pm0.04}$ | $75.00_{\pm0.03}$ | $95.00_{\pm0.07}$ | $74.73_{\pm0.02}$ |

discern that in the unbalanced noise case, the performance drop is more severe than the balanced case. Moreover, we made three observations in Fig. 11 that are noteworthy; these observations are reminiscent of the observations that were made in Section 4.

(i) In Fig. 11, the diagonal represents the balanced noise injection. It can be seen that the performance consistently deteriorates when moving from top-left entry (no noise) to bottom-right entry (intense noise); if we recall, this is exactly the opposite of the observation made in Fig. 3: there, injecting balanced noise reliably and significantly increased the performance. (ii) We observe that in Fig. 11 unbalanced noise worsens the performance which is in accordance with the observation in Fig. 3. Finally, (iii) just as in Fig. 3, there is an asymmetry in performance for having an unbalanced noise: strong noise during inference with weak noise during training delivers noticeably worse performances than otherwise.

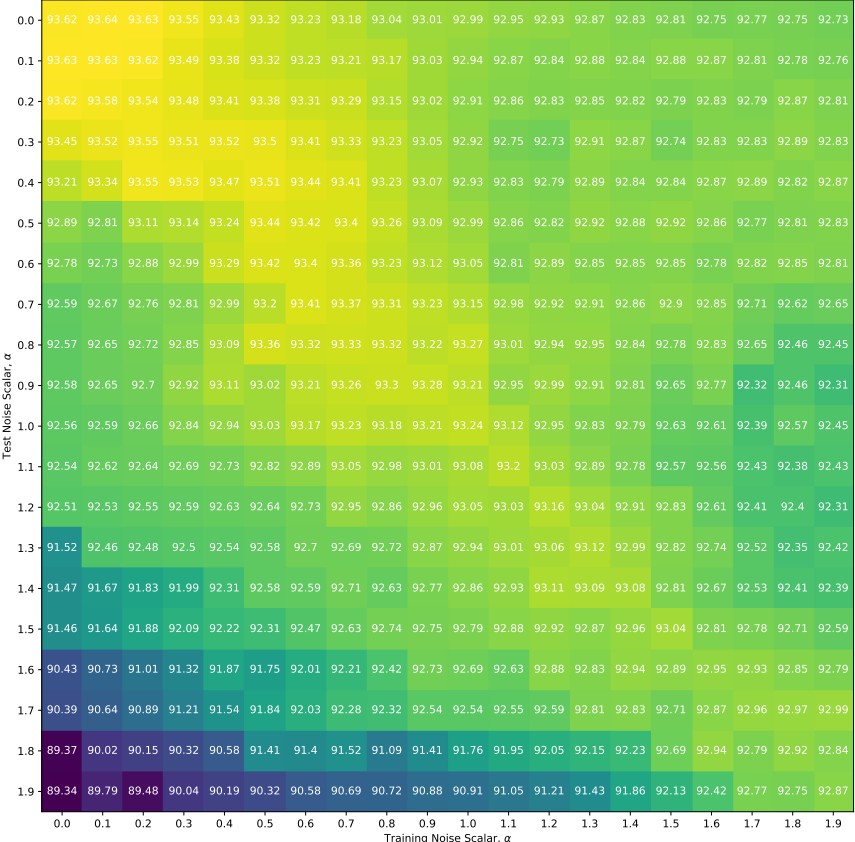

Figure 11: The classification accuracy of ResNet50 architecture on CIFAR-10 dataset as the noise scalar $\alpha$ increases both during training and inference for the standard Gaussian noise $\bar{z}$.

Our results for CIFAR-10 dataset in Figs. 10 and 11 carry over to CIFAR-100 dataset; hence, we refrain from reporting the results pertaining to CIFAR-100 in separate figures. From the results that have been reported in Figs. 10 and 11, and also in Table 5, it becomes clear that, in contrast to generative modeling, in discriminative modeling, the activation noise (injected to all-layers in both training and inference) indeed worsens the performance.

## F ON WHY ACTIVATION NOISE IS EFFECTIVE FOR GENERATIVE MODELING BUT NOT DISCRIMINATIVE MODELING

We do not have any rigorous mathematical proof showing why activation noise is effective for generative modeling but not discriminative modeling. However, we can conjecture the reason. To this end, we need to take a step back and ponder about the function of neural networks. As we know, in the broad mathematical sense, neural networks are best understood as approximating often a non-linear target function $j(\cdot)$ that maps input variable $X$ to an output variable $Y$, where $X$ and $Y$ come from training data $\mathcal{D}$. Specifically, the aim is learning the target function $j(\cdot)$ from training data $\mathcal{D}$ via having $j(\cdot)$ to perform curve-fitting on the training data $\mathcal{D}$.

For both the discriminative and generative modeling scenario, the dimensions of the input space $X$ is the same; however, when it comes to the output space $Y$ the difference emerges: in a discriminative modeling setting, the output space has far fewer dimensions than that of generative modeling. For example, in the CIFAR-10 dataset, the output space $Y$ for the generative model has $32{\times}32{=}1024$ dimensions whereas the discriminative model has only 10 dimensions. Meanwhile, we know that as soon as the number of dimensions grows, the curse of dimensionality comes to play: our training data $\mathcal{D}$ now becomes exponentially insufficient; the data becomes sparsely distributed in the space,

and instead of having a smooth continuous manifold of the data which is desired, we will have patches of data scattered in the space. This makes it hard for our $j(\cdot)$ to map the input to the output because neural networks, as hinted in Universal Approximation Theorem (Hornik et al., 1989), in the strict sense cannot, and in the practical sense, severely suffer when approximating functions whose samples in the input space are not smooth.

The activation noise, as proved in the main text where we linked activation noise with data augmentation, imputes the primary dataset $\mathcal{D}$ with an auxiliary dataset $\mathcal{D}'$ to smooth the data manifold, thereby mitigating the curse of dimensionality; however, when the output has fewer dimensions, as in the case of discriminative modeling, this problem is less pronounced. That said, one question that might arise is that these explanations delivered so far justify adopting activation noise during the training as an regularization scheme; how activation noise during inference comes to play and interact with the activation noise during training?

When we make the manifold of our dataset $\mathcal{D}$ more continuous and smooth during training by augmenting it with $\mathcal{D}'$, we are instructing the neural network to learn an enhanced manifold instead of the original one: a new one that is smoother, more continuous, and stretched. During inference we compute the likelihood of a given sample $x$ under our model; if the noise scalar is considerably larger than that of training time, it causes to stretch the cloud of our sample points; then clearly the manifold that is learned by neural networks has not seen the far points of the cloud and included them in itself during training; hence, the neural network would produce unreliable noisy likelihoods for these outlier points of the cloud. This is why the performance severely deteriorates when the noise in inference has a larger scalar than that of training (even worse than no sampling/noise). In other words, larger noise scalar $\alpha$ at inference causes the cloud of noise samples fall outside of the convex hull of the dataset on which the neural network is trained. Eventually, as we presented in Theorem 1, the sampling cloud must be of the same shape/distribution as the training cloud of noise that is used to smooth the dataset $\mathcal{D}$.

