# OpenReview forum: "ON INJECTING NOISE DURING INFERENCE"
_ICLR.cc/2023/Conference — Submitted to ICLR 2023_

### Official Review · Reviewer_BpSj · 2022-10-21

**Confidence:** 5
**Correctness:** 2
**Technical Novelty And Significance:** 1
**Empirical Novelty And Significance:** 2
**Recommendation:** 3

**Clarity, Quality, Novelty And Reproducibility:**

The paper is written clearly, easy to follow.

The novelty of the paper is under the question. For example, [1] already in 1996 considered different types of noise showing that it performs regularization. The claim of equivalence to dropout is not correct and equivalence to data noise does not require a proof. Analogously, the claim that usage of the same noise during inference and training does not require a proof. Moreover, original inference scheme of the dropout is sampling, but since it is computationally expensive the rescaling approach was proposed: thus inference with sampling is not novel as well.

The supplementary material includes code, so the results are reproducible with high probability.

[1] G.An "The effects of adding noise during backpropagation training on a generalization performance"

**Strength And Weaknesses:**

The paper demonstrates results of a very large amount of experiments, performed for different setups and with standard deviation of the run results. Unfortunately, there are no comparisons to any other regularizations and no reports of state-of-the-art results in the field, so it is very hard to judge how beneficial the proposed scheme is.

The paper is greatly over-claiming the obtained results.
The proposed proof of the equivalency of dropout to additive noise is based on the introduced in the paper negate random variable, which is supposed to follow the distribution of activations but being negative of the value. First, I am not sure that constructing such a variable possible in general. Second, this does not prove equivalence to binary dropout, because a mask is sampled anew every time - this will mean that additive noise has to change distribution on every step to correspond to the new mask. Moreover, dropout can be not only binary, but also continuous.
The announced in the abstract 200% improvement of the accuracy is seen only in the particular setup considered. It is not specified if the initial model (without noise) has been fine-tuned to perform best on the problem. In such setup adding regularization can obviously lead to very large positive changes, not confirming that particularly additive noise can improve a model twice.

**Summary Of The Paper:**

The paper investigates the effect of noise injection in the form of additive noise during training and inference of a specific class of neural network models energy based generative models, which are roughly variational encoders whose output is interpreted as energy: smaller energy corresponds to the correct class prediction. Several claims about properties of additive noise are made and multiple experimental results are provided. In particular, it is claimed that additive noise is equivalent to dropout and data augmentation, also that additive noise during inference improves accuracy. Experiments compare multiple different setups of the proposed noisy model on different datasets.

**Summary Of The Review:**

The paper investigates an interesting matter of additive noise injection during training and inference. The problem is considered in a particular setup of energy based models. The theoretical claims made in the paper are not significantly novel and some are incorrect. The practical results are presenting an extensive evaluation of the different setups of the model, without any benchmarks or comparisons.
In the current state the paper is not ready for publication.

----
I thank the authors for the reply. Nevertheless, I believe that in its current form the paper has flaws that prevent it from being published. Therefore I stick to my current score.

---

> ### Author Response · Authors · 2022-11-18
> **Our novelty is that regularization (in training) and sampling (in inference) are jointly studied.**
>
> "Unfortunately, there are ... proposed scheme is."
>
> Although in the revised manuscript we included the simulation results of dropout (See Fig. 2), the reasons for a lack of comparison in the original manuscript with other regularization schemes is that our work is not just limited to regularization, nor can be reduced to sampling. Our work investigates the joint impacts of noise during both training and inference.
>
> We know activation noise during training does regularization and we know activation noise during inference can be interpreted as sampling. Our work shows that when we do both there comes an emergent behavior that relies on both. Therefore, our work stands on a unique position which makes it difficult to compare.
>
> "The paper is ... the obtained results."
>
> We made our code available to the public for reproduction
>
> "The proposed proof ... to correspond to the new mask."
>
> Dropout is all about cancelling an activation function. Meanwhile, we can state that the noise can cancel a neuron's activation function if we know that the distribution of the noise is the negate of the signal. For that the distribution of the noise must be a function of the activation function (a negate of it) which is doable and, hence, "a mask does not need to be sampled anew every time" and surely "constructing such a variable possible in general."
>
> "The announced in the ... the problem."
>
> The reported 200% increase in accuracy was observed in a fair experiment (that can be reproduced). The baseline is when our network is injected with no noise during training (with 100 epochs) nor during inference which yields the performance of 20.48%. Hence, yes, indeed "the initial model (without noise) has been fine-tuned to perform best on the problem."
>
> When we inject noise with the amplitude of 0.9 during both training and inference, we observe the performance of 64.55%. The same network, the same number of epochs, and the same optimizer, but totally different outcome.
>
> The statement that "adding regularization can obviously lead to very large positive changes, not confirming that particularly additive noise can improve a model twice" is irrelevant to our work. If the statement of the reviewer were true, then we would see the maximum performance on the first row of Fig. 3 which corresponds to the regularization.
>
> Again, our work is not just a matter of regularization. The observed increase in performance cannot be only attributed to either of the regularization or sampling as it becomes clear by observing the diagonal of Fig. 3. And it is worth noting that the announced 200% improvement of the accuracy is {\it not} seen only in one setup; we observe similar performance increases for all four other datasets considered in our paper.
>
> "The novelty of the paper is … performs regularization."
>
> The novelty of our work {\it would be} under question, had it been simply about regularization via activation noise. However, what we discover is a much more nuanced phenomenon. Our work studies the relationship between activation noise during training and inference. Therefore, the novelty of our work is not under question.
>
> Fig. 3 testifies for our statement: we see that the magic happens not just when there is regularization or when there is sampling but when both are present. The paper [1] in 1996 only considered the role of noise as a regularizer. No sampling. And no discussion about the relationship between regularization and sampling. Also, no extensive experiments with many datasets. Therefore, the novelty of our work remains firmly valid.
>
> "Moreover, original ... novel as well."
>
> Even though the original inference scheme of dropout is Monte-Carlo model averaging via sampling, the original dropout paper [1] neither does discuss nor provide any evidence as to what the distribution of the sampler should be compared to the distribution of the activation noise during training.
>
> Our work pushes the conversation one step further: we propose activation noise which is a general form of dropout and can act as dropout. We discuss what should be the relationship between the injected activation noise during training and inference.
>
> In Fig. 3, the amplitude of the activation noise must match during training and inference; and that can make the performance 3x better or worse. No work in the literature so far has discovered this. And, our work, compared to the original dropout work [2], offers ample empirical results for many datasets.
>
> Also, dropout is not as effective for convolutional neural networks (see Fig. 2). This has been noted in the literature [3]. In contrast, activation noise, a general form of dropout, exhibits broader applicability (see Figs. 2 and 3 as well as Tab. 1).
>
> [2]: Dropout: A Simple Way to Prevent Neural Networks from Overfitting
> [3]: Towards Dropout Training for Convolutional Neural Networks

---

> > ### Comment · Reviewer_BpSj · 2022-11-24
> > **Reply**
> >
> > I thank the authors for the clarifications.
> >
> > I still would insist on the following points:
> >
> > 1 - Dropout as multiplicative noise cannot be equivalent to an additive noise. Additive noise indeed will have to be a function of an activation in order to be similar to a multiplicative noise. Since in a realistic case we cannot describe the distribution of activations, producing such noise is not an easy task. Moreover, on every training step, with changing the parameters of the model, the distribution of activations is changing, thus requiring change of the distribution that is a function of it. Moreover,
> > 2 - It is possible, that usage of sampling in the inference when additive noise is used has not been widely researched. At the same moment, the sampling in the inference with dropout is the way to go (check out https://www.deeplearningbook.org/ chapter7.12).
> > 3 - In order to support the claim about 200% improvement it is needed to consider a rather wide range of good performing neural networks, whose performance will be improved via adding noise while training and inference.

---

> > > ### Author Response · Authors · 2022-11-25
> > > **Authors' Reply (1/2)**
> > >
> > > 0. Regarding the comment “I thank the authors for the clarifications.”
> > >
> > > Thank you very much for your reply. If the Reviewer has any further comments or concerns on our main novelty and contribution, which is joint study of regularization (training) and sampling (inference), we would be happy to further clarify the issue.
> > >
> > > 1. Regarding comment “Dropout as multiplicative noise cannot be equivalent to an additive noise...”
> > >
> > > In our paper, we never claimed that activation noise is equivalent to dropout. We clearly and explicitly said that the former {\it can} be considered as {\it a general} form of the latter, or the latter {\it can} be considered as {\it a special} case of the former. Please see the sentence right below Proposition 1 in our paper, which reads:
> > >
> > > "This theoretical result implies that activation noise can be considered as a general form of dropout."
> > >
> > > In our paper, what we claimed was this: Dropout cancels a signal at a neuron with a pre-determined non-zero probability (via multiplication by zero), and our activation function can also cancel a signal with a non-zero probability density (when the noise becomes the negate of the signal). It is very clear that the two approaches are not equivalent. In our paper, we only claimed that the activation function can be considered as a general form of dropout.
> > >
> > > Symmetric noises like Gaussian noise can be considered in principle as a general case of dropout; in other words, symmetric noises can in principle cancel signals, which is what dropout does in the end of the day (but, by multiplying zero). This is because the range of symmetric noises includes the values that can negate the activation function although it is not desired.
> > >
> > > It is important to note that although it is in principle possible, we never intend to arrive at dropout via activation noise; rather, our aim is to go well beyond dropout and come up with activation function enhancement schemes that outperform signal cancelling (as dropout). Specifically, we did not want to enforce that the signal is completely canceled by the activation function; instead, the activation function works essentially as regularization and data augmentation in practice.
> > >
> > > As can be seen in Figure 2 of the revised paper, the activation noise, as it is, significantly outperforms dropout for two datasets. If activation noise and dropout {\it were} equivalent, their performances {\it would} be the same. However, as shown in Figure 2, activation noise performs much better than dropout. This means that activation noise is something more than dropout. Also, activation noise shows unprecedented performance in Figures 3 and 4 as well as Tables 1 and 2 for five datasets. All in all, directly turning the activation noise into dropout is not intended (nor desired) in our paper. We just established their relationship through Definition 1 and Proposition 1: activation noise can be considered as a general form of dropout.
> > >
> > > 2.1 Regarding comment "It is possible, that usage of sampling in the inference when additive noise is used has not been widely researched."
> > >
> > > Thank you very much for this comment. What the Reviewer commented is indeed one of our major contributions, which fills the gap in the literature. We know activation noise during training does regularization and we know activation noise during inference can be interpreted as sampling. Our work shows that when we do both, there comes an emergent behavior that relies on both of them. In other words, if any of the above two are absent then we will not observe the radical increase in the performance. Therefore, our work stands in a unique position in the literature.
> > >
> > > 2.2 Regarding comment “At the same moment, the sampling in the inference with dropout is the way to go (check out https://www.deeplearningbook.org/ chapter7.12).”
> > >
> > > That is exactly what we did and the result was added to the revised manuscript during Discussion Stage 1. Specifically, in Figure 2, we added new simulation results for “the sampling in the inference with dropout (in training)” and we compared its performance with our scheme of sampling in the inference with activation noise (in training). As can be seen, our approach works much better, which confirms all our theoretical results and discussions.
> > >
> > > In the end, note that dropout [1] has limitations. Dropout is not as effective for convolutional neural networks as it is for dense neural networks [2]. Even for dense neural networks, it is applied at certain layers. Still, it barely increases the performance by 1% if any. Now compare that with the significant accuracy increase observed in activation noise (e.g., from 50% up to 200% for five datasets). Please see Tables 1 and 2 as well as Figures 2 and 3.

---

> > > > ### Author Response · Authors · 2022-11-25
> > > > **Authors' Reply (2/2)**
> > > >
> > > > 3. Regarding comment “...it is needed to consider a rather wide range of good performing neural networks...”
> > > >
> > > > For the broad applicability of the activation noise, our activation noise as it has been shown is beneficial for convolutional neural networks. We also have the results indicating that activation noise is impactful for dense neural networks as well. In contrast to this, dropout [1] has not shown as much generality because its results are not as promising for convolutional neural networks [2]. Nor dropout could compete with our activation noise when the same sampling strategy (with the same distribution) was used in inference; see Figure 2.
> > > >
> > > > We have the results supporting the effectiveness of the activation noise for different sizes of convolutional neural networks and different sizes of dense neural networks. Thus, we are convinced that activation noise is applicable to a wide range of neural networks (and datasets). Figures 5 and 6 present our results for when different activation noises to different layers of the neural networks are applied.
> > > >
> > > > Regarding the amount of performance improvement, we observe about 200%, 100%, 50%, 50%, and 50% improvements for CIFAR-10, CIFAR-20, CalTech-20, Flower-20, and CUB-20 datasets, respectively.
> > > >
> > > >
> > > > [1]: Dropout: A Simple Way to Prevent Neural Networks from Overfitting
> > > > [2]: Towards Dropout Training for Convolutional Neural Networks

---

### Official Review · Reviewer_Zh4D · 2022-10-25

**Confidence:** 3
**Correctness:** 3
**Technical Novelty And Significance:** 3
**Empirical Novelty And Significance:** 2
**Recommendation:** 6

**Clarity, Quality, Novelty And Reproducibility:**

Barring a few minor issues, the paper is easy to read and the insights look novel. No issues on reproducibility.

**Strength And Weaknesses:**

Strengths
- This paper generalizes dropout to activation noise and studies it thoroughly using controlled set of experiments
- Using activation noise at inference time seems novel (I have not seen it before) and likely generally useful
- The paper also presents an interesting negative result that activation noise is not effective for discriminative modeling.

Weakness
- Caption for Table 1 can be made more descriptive so that the reader can look at it in a self-contained way.
- While the phenomena is novel, the most impressive results are obtained using 10^4 samples on all the datasets. I'm unsure as a reader if this has practical benefit even on a moderately larger models than the paper considers. I think at least a discussion how the insights can be used would make the paper stronger.

**Summary Of The Paper:**

The paper generalizes and studies activation noise for both training and inference for the specific case of energy-based models. Core contributions include 1) using activation noise helping at inference time (typically dropout is not used at inference time) and 2) through study of which pairs of distributions of noise works best at training and inference time

**Summary Of The Review:**

While the paper presents interesting observations using thorough experiments, the practical significance of these results is not apparent.

---

> ### Author Response · Authors · 2022-11-18
> **Authors' Replies**
>
> We appreciate the reviewer's encouragements. The reviewer's words mean a lot to us.
>
> We improved the caption for Table 1 in the revised manuscript. The new caption is more descriptive and provides the information in a more self-contained manner.
>
> "While the phenomena is novel, the most impressive results are obtained using 10^4 samples on all the datasets. I'm unsure as a reader if this has practical benefit even on moderately larger models than the paper considers."
>
> The sample size does not inflict any issue because as it can be seen that even with sample size of 10 the model with activation noise yields remarkable results which is 1.7x better compared to sample size of 1. For that please see Table 2 where for the noise values of (0.9, 0.9), we have the following results (25.25, 42.04, 53.02, 64.55, 65.88) pertaining to the sample sizes of (1, 10, 10^2, 10^3, 10^4). Furthermore, all different samples of our model with activation noise can be easily calculated, allowing us to have all the samples computed at once because the inference is highly parallelizable. Hence, the inference phase would not experience any issue due to the sample size.
>
> "I think at least a discussion how the insights can be used would make the paper stronger."
>
> The main insight is that activation noise during training and inference can radically improve the performance of generative classifiers which are applicable to the problems of density estimation. Accordingly, in this paper instead of learning the conditional probability p (y | x) of the dataset which is what usually is being done in discriminative modeling, the joint probability p (x, y) of the dataset is learned. This task is of importance in Out-of-Distribution Detection, Anomaly Detection, One-Class Classification, Lifelong and Continual Learning, and Federated Learning.
>
> In Out-of-Distribution Detection and Anomaly Detection applications, usually instead of learning different classes via discriminative modeling p (y | x), we tend to learn different classes by generative modeling p (x, y). This enables us to have the probability density of each class which helps to identify if any given new sample falls inside or outside of the distribution that our network is trained on. If the sample falls outside of the intended distribution, then it is called an anomaly.
>
> In One-Class Classification, we again learn different classes via generative modeling because the fundamental assumption is that there is only one class; hence, it is not possible to adopt the common discriminative modeling approaches that require to be fed by at least two classes.
>
> We highly recommend that the reviewer see the following popular paper where the authors adopt a similar task to ours for Out-of-Distribution Detection and Anomaly Detection.
>
> "Deep One-Class Classification," PMLR 2018
>
> Furthermore, recently, this task has found its way into lifelong and continual learning applications where expansion-based architectures and mixture models are becoming increasingly relevant. Please see the following paper where the authors adopt generative classifiers instead of discriminators to tackle the problem of catastrophic forgetting in lifelong and continual learning applications.
>
> "Class-Incremental Learning with Generative Classifiers," CVPR 2021
>
> We expect this task to be of interest to Federated Learning because of the importance of learning in a one-class fashion which is faced in scenarios where each client in Federated Learning possesses only one class.
>
> Furthermore, there is a limitation to dropout: dropout is not as effective for convolutional neural networks as activation noise is as it can be seen in Fig. 2 of the revised manuscript. This has been noted in the literature such as in [1]. Whereas, activation noise, a general form of dropout, exhibits broader applicability (see Figs. 2 and 3 as well as Tab. 1).
>
> Please see Section 2 of literature review in the revised paper for more discussions.
>
> [1]: Towards Dropout Training for Convolutional Neural Networks

---

### Official Review · Reviewer_Ysne · 2022-10-25

**Confidence:** 2
**Correctness:** 2
**Technical Novelty And Significance:** 2
**Empirical Novelty And Significance:** 1
**Recommendation:** 3

**Clarity, Quality, Novelty And Reproducibility:**

The paper has some clarity and quality concerns based on the comments above. There is some novelty in the sense that adding noise to EBM is a new idea. The paper has code in the supplementary file.

**Strength And Weaknesses:**

Strengths:

- The paper has a good overview of literature.
- Adding noise to EBM seems to be a novel approach.
- Experiments show huge improvement on classification results.

Weaknesses:

- The title is too big and does not match the paper. Noise is added in both training and inference. Also, since the paper focuses on EBM applied to classification tasks, the title must reflect that.
- The presentation of methodology is confusing.
  - It would be nice to have a detailed paragraph on how to mathematically perform classification with EBM (or ten AEs for these classes). I notice that you perform $\arg\min_y E(x|y)$ in the paragraph after eq(10). Is it just intuition or derived from Bayes rule applied to $p(y|x)$?
  - Def 1 and Prop 1 seems straightforward and do not convey much insights. In practice we do not expect the dropout case to happen. What theory do you have for other types of noises?
  - Eq (5): why use MSE for energy? Why can you decompose $x$ with independent $x_k$?
  - Thm 1 should be in methodology section rather than experiment section. A theorem concluding empirical findings is much weaker than one that predicts. In addition, this theorem seems straightforward and I do not find anything interesting.
- The experiments do not show interesting results.
  - How is the task different from standard classification tasks? If not, why are we interested in this approach, instead of just using standard methods?
  - What are some interesting applications of the proposed method?


**Summary Of The Paper:**

The authors propose a method to add noise in EBM in classification tasks. The likelihood and loss functions are derived based on the injected noise $z$. The paper claims adding noise during training includes dropout as a special case, and propose to add the same amount of noise during inference. On image classification tasks, the paper shows increased accuracy when adding the proper amount of noise.

**Summary Of The Review:**

I think the paper needs significant improvement on writing and technical results. I do not think this draft can be accepted, but my decision might change based on authors' reply and revised version.

---

> ### Author Response · Authors · 2022-11-18
> **Authors' Replies**
>
> "The title is too ... must reflect that."
>
> The new title is "Activation Noise for EBM Classification."
>
> "It would be nice ... applied to?"
>
> Please see Section 3.2.
>
> "Def 1 and Prop 1 seems ... types of noises?"
>
> Def. 1 and Prop. 1 help to clarify the relationship between our activation noise and dropout. However, we do not intend our activation noise to become dropout; our goal is rather to go beyond dropout. As the reviewer mentioned, "we do not expect the dropout case to happen," nor we desire our activation noise to reduce to dropout. Although not desired, it is still possible for activation noise to act like dropout if the activation noise distribution follows the negate distribution.
>
> For the question "what theory do you have for other types of noises?" we can say that in general, any symmetric noise distribution like Gaussian, Cauchy, uniform, Chi, and t-Distribution can negate the activation function and are a general form of dropout.
>
> Also, we are the first to discover and prove that the distribution of activation noise in the inference (for sampling) must match the distribution of activation noise during training.
>
> In Fig. 3, the amplitude of the activation noise must match during training and inference; and that can make the performance 3x better or worse. No work has discovered this. Moreover, our work, compared to the original dropout work [1], offers ample empirical results for many datasets. Please see Tab. 1 and the Appendix.
>
> "Eq (5): why use MSE for energy?"
>
> Among all the available alternatives like binary cross entropy, we found MSE to deliver the best results in practice with more reliability. Therefore, we based our discussions on MSE.
>
> "Why can you decompose x with independent x_k?"
>
> In the MSE equation of (5), x is not a random vector but a deterministic one. Note that there is no expectation in (5). The squared l2 norm of a deterministic vector is the summation of square of all elements, x_k. Also, please see the following Wikipedia page.
>
> https://en.wikipedia.org/wiki/Mean_squared_error
>
> "Thm 1 should be ... experiment section."
>
> We believe that Theorem 1 is placed in the right section. We placed Theorem 1 in "Theories for the simulation results" section which comes after the preliminary experiments to discuss the observations.
>
> "A theorem concluding … anything interesting."
>
> Theorem 1 does actually predict the extensive simulation results that come in the next section entitled "Comprehensive simulation results for Gaussian Noise" and also those results that come in the following section "Integrating other noise distributions."
>
> Moreover, we are the first to prove that the distribution of activation noise in the inference must match the distribution of activation noise during training. Furthermore, our work, compared to the original dropout work [1], offers ample empirical results for many datasets. Please see Table 1 and the Appendix.
>
> "The experiments do not show interesting results."
>
> Our experiments do show very interesting and meaningful results. In Fig. 3, we see that just by injecting noise we can radically increase the accuracy by 3x (or 200%). We find that the distribution of noise during training and inference must match, which has never been discovered. We recommend that the reviewer reproduce the Fig. 3.
>
> We know activation noise during training does regularization and we know activation noise during inference can be interpreted as sampling. Our experiments show that when we do both there comes an emergent behavior that relies on both of them. In other words, if any of the above two are absent then we will not observe the radical increase in the performance.
>
> "How is the ... standard methods?"
>
> "What are ... proposed method?"
>
> In Out-of-Distribution Detection and Anomaly Detection, instead of learning different classes via discriminative modeling p (y | x), we tend to learn different classes by generative modeling p (x, y). This enables us to have the probability density of each class which helps to identify if any given new sample falls inside or outside of the distribution that our network is trained on.
>
> In One-Class Classification, we again learn different classes via generative modeling because the fundamental assumption is that there is only one class; hence, it is not possible to adopt the common discriminative modeling approaches that require at least two classes.
>
> See the following popular paper where the authors adopt a similar task to ours.
>
> Deep One-Class Classification, PMLR 2018
>
> Please see the following paper where the authors adopt generative classifiers instead of discriminators to tackle the problem of catastrophic forgetting.
>
> Class-Incremental Learning with Generative Classifiers, CVPR 2021
>
> Also, dropout is not as effective for CNNs (see Fig. 2). This has been noted in the literature [1]. In contrast, activation noise, a general form of dropout, exhibits broader applicability (CNNs).
>
> [1]: Towards Dropout Training for Convolutional Neural Networks

---

> > ### Author Response · Authors · 2022-11-25
> > **Further clarification to the comment: "Why can you decompose x with independent x_k?"**
> >
> > For this comment, we’d like to further clarify our previous answer: “...x is not a random vector but a deterministic one. Note that there is no expectation in (5).” In this answer, we meant that the vector x is deterministic given the joint PDF p_D(x, y). We also meant that we did not decompose the expectation over x into a simple multiplication of expectations over x_k. Note that in Equation 5 and all subsequent equations, we never turn the joint PDF p_D (x, y) into a simple multiplication of its marginal PDFs p_D(x_k, y). If we had done it, the condition of independence would have been needed. In our derivation, we do not decompose the joint PDF p_D(x, y) into its marginal PDFs; instead, we carry the joint PDF as is up to the very last equation. Hence, keeping P_D(x, y),  the decomposition of x into x_k is warranted/valid.
> >
> > In fact, our analysis in equations (5)-(8) is still valid even without the decomposition of x into x_k. The reason why we decompose x into x_k (given P_D(x, y)) is to better present the equations in the form of MSE loss, which is typically computed at the pixel level. The conclusion we derived for the MSE can be extended for other loss functions. In other words, not only for MSE as we have done, but also for any arbitrary loss function, we can re-write the loss as a combination of the losses pertaining to the signal component plus the noise component, and then the same result would hold. All in all, we are even not necessarily required to decompose x into x_k (given P_D(x, y)).

---

### Official Review · Reviewer_qks6 · 2022-10-26

**Confidence:** 4
**Correctness:** 3
**Technical Novelty And Significance:** 3
**Empirical Novelty And Significance:** 3
**Recommendation:** 5

**Clarity, Quality, Novelty And Reproducibility:**

The paper is clearly written.
The paper is novel as far as I can tell.
I believe the paper is reproducible.

**Strength And Weaknesses:**

**Strengths:**
* I believe the topic is important and the community can benefit from papers taking a theoretical look at regularisation methods like this.
* The paper is well written and included some interesting insights.
* I appreciate the experiments done by the authors, especially that the benefits of noise during inference are investigated.


**Weaknesses:**
* Unfortunately, some of the theory is not completely clear to me and I am not sure I agree. See below.
* The authors say that that dropout is a special case, but they do not include it in their experiments. This is unfortunate, because dropout is so widely used
* I think the authors miss a point about the connection between dropout and activation noise.

**Detailed comments:**
Most importantly, I think the formulation in Eq. 4 might be problematic.
The authors give their objective for jointly optimise the network parameters theta and noise distribution q(z) to maximise the conditional log likelihood of their training data drawn form the distribution p_D(x,y).
However, their model p_theta(x|y,z) is conditioned on the noise z instead of integrating over it.
This objective essentially tries to make p_theta(x|y,z) approximate p_D(x|y) (which could be derived from p_D(x,y)) for each z sampled form q(z).
Considering now that p_D(x|y) does not depend on z, the optimal solution for this objective would be to make q(z) infinitley concentrated on 0, as z cannot contribute to better approximate p_D(x|y), right?

Later in Eq. 9, when the authors discuss inference, they use a different formulation, integrating the energy over the noise and then choosing the class with the lowest integrated energy. Here, the noise becomes part of the model.
I think this approach is more sensible, but it is in conflict with Eq. 4, where we are integrating the log probabilities.
This is different, because of the partition function, right?

Regarding viewing dropout as special case of activation noise, I agree with this perspective.
However, in contrast to the perspective taken by the authors, the activation noise distribution that would result in dropout behaviour, would have to depend on the input x, as it has to (with a certain probability) exactly cancel out the activation of the neuron.
The types of activation noise considered in the paper generally don't depend on x.
Does the finding that we should use noise during inference hold for dropout?










**Summary Of The Paper:**

The authors investigate the effect of activation noise (additive noise applied after the activation function) on an EBM based classification approach.
They mathematically describe activation noise during training and inference and come to the conclusion that (i) activation noise can be understood as a generalisation of dropout and (ii) that the loss at inference is minimised when the distribution of noise is used during training and inference.
In their experiments, the  authors demonstrate the latter point (ii) empirically and compare the effect of different activation noise distributions on classification accuracy.
They find that symmetric noise distributions are to be preferred.



**Summary Of The Review:**

All-in-all, I think this is an interesting paper on an important topic.
However, there are parts of the theory that I cannot follow.
I believe this has to be clarified to be ready for publication.

---

> ### Author Response · Authors · 2022-11-18
> **Authors’ replies**
>
> "the optimal solution for this objective would be to make q(z) infinitely concentrated on 0, as z cannot contribute to better approximate p_D(x|y), right?"
>
> This intuition appears correct, theoretically. However, surprisingly, in practice reality is in contrast with the intuition and our experiments tell a different story: the empirical results all confirm the hypothesis that actually there is a non-zero q(z) that helps p_theta(x|y,z) to approximate p_D(x|y) more effectively. There is one of the contributions of this paper which promises the existence of a noise distribution q(z) that not only does not hurt our goal of maximizing the conditional loglikelihood of p_D(x|y) but also facilitates it, remarkably.
>
> The reason for this surprising result is that in practice commonly we do not solve (analytically or directly) the integration in Eq. 4 hidden in the expectation symbol and also the integration in Eq. 9; instead, we turn the integration into a summation that operates over finite data samples (as opposed to infinite) and solve the problem of Eq. 4 and Eq. 9 numerically (discretely) as opposed to theoretically (continuously). When we inject the noise to the neural networks in training, it can be beneficial as we discussed in Section 3.1: Activation noise in training is related to dropout, loss regularization, and data augmentation. Taking the view of data augmentation as an example, adding activation noise in training is as if we multiply the data samples; therefore, the numerical answer would better approximate the theoretical one both for Eq. 4 and Eq. 9. For the views of dropout and loss regularization, please see Section 3.1.
>
> "Later in Eq. 9, when the authors discuss inference, they use a different formulation, integrating the energy over the noise and then choosing the class with the lowest integrated energy. Here, the noise becomes part of the model. I think this approach is more sensible, but it is in conflict with Eq. 4, where we are integrating the log probabilities. This is different, because of the partition function, right?"
>
> As the reviewer noted, our formulations in Eqs. 4 and 9 pertaining to training and inference differ. This, as the reviewer suggested, is due to the reason that the partition function intermediates between energy function and the loglikelihood; and the fact that we make the inference based upon the energy function which is different from the training where we minimize the loglikelihood in the optimization function. Nevertheless, that does not cause an issue in the formulations. It is just that in the training we minimize the loglikelihood while the inference is based upon the energy function. Totally normal.
>
> "Regarding viewing dropout as special case of activation noise, I agree with this perspective. However, in contrast to the perspective taken by the authors, the activation noise distribution that would result in dropout behaviour, would have to depend on the input x, as it has to (with a certain probability) exactly cancel out the activation of the neuron. The types of activation noise considered in the paper generally don't depend on x. Does the finding that we should use noise during inference hold for dropout?"
>
> As the reviewer pointed out, "the activation noise distribution that would result in dropout behaviour, would have to depend on the input x," in order to cancel out the activation of a neuron. And ``the types of activation noise considered in the paper generally don't depend on x." The reason is that although it is possible, we did not specifically intend to arrive at dropout via activation noise; our aim rather is to go beyond dropout and come up with activation function enhancement schemes that outperform dropout. Therefore, we did not want to make sure that the activation function is canceled; instead, we planned to have the activation function augmented by noise.
>
> Interestingly, symmetric noises like Gaussian noise are in principle a general case of dropout; in other words, it can be shown that symmetric noises can cancel the activation function and do what dropout does. This is because the range of symmetric noises include values that in principle can negate the activation function although it is not desired.
>
> To answer the question that "does the finding that we should use noise during inference hold for dropout?" from our theoretical results we deduce that the answer is yes which is in agreement with what we know in the literature [1]. Indeed, in the original dropout paper this idea was discussed, and the authors recommended the adoption of sampling for dropout during inference.
>
> In the end, to address the reviewer's comment, our revised paper includes experiments presenting results pertaining to dropout for CIFAR-10 and CIFAR-20 and contrasts them with the results of noise. It can be seen that the noise scheme outperforms dropout. Please see Fig. 2.
>
> [1]: "Dropout: A Simple Way to Prevent Neural Networks from Overfitting"

---

> > ### Comment · Reviewer_qks6 · 2022-12-02
> > **I am unfortunately not convinced.**
> >
> > Unfortunately, I am not convinced by the arguments.
> > Regarding the fact that the optimal distribution for q should be concentrated around 0:
> > I understand your argument as saying that the optimal solution should be concentrated at 0, but that you particular implementation does not find this solution, but one that is actually preferable. If this is the case, I find it has to be theoretically explored why.
> > Surely, adding the noise can only be preferable because it helps with generalisation, not because it helps to fit for the training data itself.
> >
> > Regarding Eq. 9:
> > You argue, that it is fine and normal to select the class with the highest energy, disregarding the normalisation.
> > However, the energy function and the normalisation function, both depend on on the class label y.
> > The same likelihood function for a class y could be represented by different scaled versions of the same energy, e.g., by multiplying a factor 10 to the energy. This would not effect the likelihood.
> > However, when you compare the different energies between different classes this would make a difference.
> >
> > I have to stick with my rating.

---

> > > ### Author Response · Authors · 2022-12-06
> > > **Authors' reply (1/2)**
> > >
> > > 0. General Answer:
> > >
> > > We would like to express our concern (and frustration). We feel that the totality of our contributions (which are novel, significant, and relevant) are simply ignored; and instead, some minor, local, and unimportant matters that in this paper we never claim to be our own contributions are highlighted. Our paper offers two main contributions: jointly studying the activation noise in training and inference plus proving the optimal distribution for inference.
> > >
> > > 1. Answer to the first comment:
> > >
> > > First of all, we’d like to clarify that our approach showed truly excellent performance improvement and this result itself is not questioned by the Reviewer. That is, we believe that the Reviewer anyway accepts that our approach improves the performance significantly, which can be easily seen from all those numerical results. If not, we invite the Reviewer (or anybody else) to run the simulations to reproduce all the results using the source codes that we have opened to the public. The point is that our approach improves the performance significantly anyway.
> > >
> > > Then, the issue raised by the Reviewer is only about the issue of {\it interpretation} for the reason why the performance is so good. We believe that even if there {\it were} no good interpretation for `why’, when the performance improvement is truly excellent, then the contributions are there anyway. This has happened in numerous works: some approaches showing excellent results are first discovered, and then, the detailed reason why is discovered or understood later.
> > >
> > > In our work, we actually have good explanation on why our approach improves the performance so much. Indeed, in the first round of the comment, when the Reviewer asked about intuitive explanation of {\it why}, we have provided our intuitions. Then, in this second round, the Reviewer commented “..., I find it has to be theoretically explored why.” That is, the Reviewer now asks for theoretical justification, because the Reviewer’s `intuition’ is different from ours, which might not be very fair.
> > >
> > > In the following, we will again do our best to deliver our intuition to the Reviewer. Before than that, we have to clarify that the understanding of the Reviewer was somewhat incorrect with regard to this comment. In the first round, the Reviewer commented “their model p_theta(x|y,z) is conditioned on the noise z instead of integrating over it.” However, this comment is not correct, because in our model p_theta(x|y,z) is integrated over z (through expectation over z), which can be trivially seen in Equation (4). Furthermore, the Reviewer commented “This objective essentially tries to make p_theta(x|y,z) approximate p_D(x|y)...for each z sampled from q(z).” However, this comment is not correct either. Our objective is {\it never} trying to make p_theta(x|y,z) approximate p_D(x|y) for each z. Instead, our objective is trying to make `integration of p_theta(x|y,z)’ approximate p_D(x|y). This can be clearly and trivially seen from the inference equation, Equation (9). Also, in the proof of Theorem 1, it is again very clear from the optimization for the inference phase, which is integration of p_theta(x|y,z) (never p_theta(x|y,z) itself) in Equation (10). We believe that, due to these misunderstandings, the Reviewer’s intuition has become different from ours.
> > >
> > > In our scheme, in training, the loss is given by integrating p_theta(x|y,z) over z, and then, the inference is performed by again integrating p_theta^*(x|y,z) over z. These can be trivially seen from Equations (4) and (9), respectively. In fact, this is one of our major contributions in this paper.
> > >
> > > Our optimization problems in Eqs. (4) and (9) are endeavoring to find noise distributions (q(z) for training and r(z) for inference) that are capable of calculating the integrations in Eq. (4) and also Eq. (9) (the integration pertaining to the expectation) more accurately. In other words, these integrals have to be solved discretely and numerically with finite number of samples from our given data distribution p_D(x, y), which immediately raises the question that how the sampling of p_D(x, y) should be done such that our approximation is more precise. The optimization problem in Eq. (4) attempts to formulate and articulate this matter. The optimization conveys the message that not only we need to find the weight parameters for our neural network but also the strategy with which we conduct the sampling of the p_D(x, y) (during training) because the latter impacts the precision of the calculation of the expectation.
> > >
> > > (continued)

---

> > > > ### Author Response · Authors · 2022-12-06
> > > > **Authors' reply (2/2)**
> > > >
> > > > Hence, injecting the noise to the neural networks in training can be beneficial as we discussed in Section 3.1. Furthermore, besides helping with accurate calculation of the expectation in Eq. (4), activation noise in training is related to dropout, loss regularization, and data augmentation. Taking the view of data augmentation as an example, adding activation noise in training is as if we multiply the data samples. In the end, the injection of such an activation noise consistently increases the performance for all the datasets from 50% up to 200%. All of the results including the ones in Fig. 3 can be easily reproduced by our codes opened.
> > > >
> > > > In the end, the critical point to note is that, in our work, the training and inference must not be considered separately or distinctively. As we studied in our paper, only when the inference is performed by integrating p_theta^*(x|y,z) over z with the noise distribution that is the same as in the training which integrates p_theta(x|y,z) over z, the excellent performance is obtained. That is, the training and inference must be considered as a whole, which is the main discovery of our paper.
> > > >
> > > > 2. Answer to the second comment.
> > > >
> > > > First of all, we have to clarify that, in this paper, we never intend to propose a particular classification method for generative classifiers. It is never our contribution, nor the intended point. Our contribution in this paper is to study and optimize activation noise jointly for the training and inference together for any likelihood-based generative models, which can be used for various applications including the generative classification as one example. Only as a simple means of demonstrating the performance, we used the energy function itself to predict the class index $\hat y$. This is only a preference and simulation choice; and therefore, does not undermine our contributions. That is why we have not even given an equation number for $\hat{y}=\underset{y^{\prime} \in \mathcal{Y}}{\arg \min } \ \hat E_{\boldsymbol{\theta}^*}\left(\boldsymbol{x} \mid y' \right).$ Hence, highlighting this matter is synonymous with missing the whole point of our paper and our significant contributions.
> > > >
> > > >
> > > > It is absolutely possible to use a slightly different way to predict $\hat y$ as follows:
> > > > \begin{eqnarray}
> > > > \hat y = \arg \min_{y’ \in \cal Y} \int_{\boldsymbol{z}} p_{\boldsymbol{\theta^*}} ( \boldsymbol{x} | y', \boldsymbol{z})  r(\boldsymbol{z}) d \boldsymbol{z} \nonumber
> > > > \end{eqnarray}
> > > > where $p_{\boldsymbol{\theta}} ( \boldsymbol{x} | y, \boldsymbol{z}) $ is given by Equation (3) with trained parameters $\boldsymbol{\theta^*}$. From Equation (3), note that $p_{\boldsymbol{\theta}} ( \boldsymbol{x} | y, \boldsymbol{z})$ is actually the normalized energy function. This trivial modification immediately and perfectly addresses the Reviewer’s comment without any changes of any other parts of the paper.
> > > >
> > > >
> > > > Clearly, whether using Eq. (9) itself or the above modified equation has essentially nothing to do with our contributions. Regardless of it, all of our contributions including Eqs. (1)--(8) and Theorem 1, are valid and concrete. In particular, in the proof of Theorem 1, we actually used the above modified equation for inference; that is, the inference is performed on $p_{\boldsymbol{\theta^*}} ( \boldsymbol{x} | y, \boldsymbol{z})$, which is the normalized version of the energy; this is exactly what the Reviewer comments.
> > > >
> > > > Because whether using Eq. (9) itself or the above modified equation  has essentially nothing to do with our contributions, and we never claim that it is any part of our own contributions, we just used the energy function (rather than the normalized one) in Eq. (9). In fact, we have also run simulations using the above modified equation, which you can find in the source codes that we have opened to the public. In the simulations, we did not observe any performance difference; we invite the Reviewer to run the simulations. The reason for observing no performance difference is that, in practice, the classification method choice (i.e., the normalization at the output or not) does not make any meaningful difference due to the existence of batchnorm at most layers.  Thus, for better (and easy) presentation of our own novel contributions, we have used the energy function for simplicity.
> > > >
> > > > All of our results and contributions whether empirical or theoretical, are all valid regardless of the classification choice. This is because the reported performance improvements were determined by comparing the behavior of the network in two scenarios: with and without activation noise. Also, note that we investigated the joint optimal activation noise for many different values of the noise scalar $\alpha$.
> > > >
> > > > All in all, if desired, Eq. (9) can be trivially replaced by the above new equation, which immediately and perfectly addresses the Reviewer's comment, without any change of our analysis or contributions.

---

> > > > ### Comment · Reviewer_qks6 · 2022-12-07
> > > > **I hope we can clarify this.**
> > > >
> > > > I am afraid I still cannot follow this line of argument.
> > > > I will try to restate my understanding below, hoping that this can help to clear up the apparent misunderstanding.
> > > > I would also appreciate hearing the other reviewers opinions on this.
> > > >
> > > > "
> > > > In the first round, the Reviewer commented “their model p_theta(x|y,z) is conditioned on the noise z instead of integrating over it.” However, this comment is not correct, because in our model p_theta(x|y,z) is integrated over z (through expectation over z), which can be trivially seen in Equation (4)
> > > > "
> > > >
> > > > I am afraid I cannot agree with this statement.
> > > > The model p_theta(x|y,z) is conditioned on z, as can be seen by considering that 'z' is written after '|'.
> > > > This model describes the probability of x given y and the noise z parametised by theta.
> > > > It is a fact that x and z are conditionally independent given y, i.e. the noise z does not carry additional information about x.
> > > > If the authors agree with me on this, it follows, that p(x|y,z) = p(x|y), i.e. the optimal solution for theta should ignore z as z cannot contribute in any way to better approximate p(x|y,z).
> > > > If the solution that is found does not lead to ignoring/removing z, than there is a mismatch between the theory and the results, i.e., the model is not really trying to approximate the real p(x|y,z).
> > > > If this is the case, why do we then bother to say that p_theta(x|y,z) should be interpreted as probability distribution at all?
> > > >
> > > > "
> > > > “This objective essentially tries to make p_theta(x|y,z) approximate p_D(x|y)...for each z sampled from q(z).” However, this comment is not correct either. Our objective is {\it never} trying to make p_theta(x|y,z) approximate p_D(x|y) for each z. Instead, our objective is trying to make `integration of p_theta(x|y,z)’ approximate p_D(x|y).
> > > > "
> > > > Eq. 4 clearly applies the logarithm inside of the integral/expectation over z not outside.
> > > > If we would want to use p(x|y,z) to try approximate p_D(x|y), we would have to first marginalise over z and then aply the logarithm as:
> > > >  -\log \int p(x|y,z) q(z) dz
> > > >  -\log p(x|y)
> > > >
> > > > "
> > > > This can be clearly and trivially seen from the inference equation, Equation (9). Also, in the proof of Theorem 1, it is again very clear from the optimization for the inference phase, which is integration of p_theta(x|y,z) (never p_theta(x|y,z) itself) in Equation (10).
> > > > "
> > > > The model training is described in Eq. 4 not Eq. 9, right? During training the approach does try to make p_theta(x|y,z) approximate p_D(x|y) for all z sampled from q. I am not sure, why the authors mention inference here.

---

> > > > > ### Author Response · Authors · 2022-12-07
> > > > > **Further clarifications on the difference between the probability and its expectation**
> > > > >
> > > > > We thank the Reviewer for the reply. We believe the second comment (on energy normalization) has been addressed. So, we will focus on the first comment, which is only an issue of ``intuitive" explanation of such huge performance improvement, not the performance itself.
> > > > >
> > > > >
> > > > > We believe the Reviewer conflates the expectation $E_{\boldsymbol{z}} \left[ p_{\boldsymbol{\theta}}(\boldsymbol{x} \mid y, \boldsymbol{z}) \right]$ with the probability $p_{\boldsymbol{\theta}}(\boldsymbol{x} \mid y, \boldsymbol{z})$, where they are related by
> > > > > \begin{align}
> > > > > E_{\boldsymbol{z}} \left[ p_{\boldsymbol{\theta}}(\boldsymbol{x} \mid y, \boldsymbol{z}) \right] = \int_{\boldsymbol{z}}  p_{\boldsymbol{\theta}} \left(\boldsymbol{x} \mid y', \boldsymbol{z} \right) q(\boldsymbol{z}) d\boldsymbol{z}
> > > > > \end{align}
> > > > > Specifically, the Reviewer seems to (incorrectly) assume that the objective of our training is to make  $p_{\boldsymbol{\theta}} ( \boldsymbol{x} | y, \boldsymbol{z})$ for every single realization of $\boldsymbol{z}$  approximate the data distribution. The incorrect parts are both "$p_{\boldsymbol{\theta}} ( \boldsymbol{x} | y, \boldsymbol{z})$"  and "for every single realization of $\boldsymbol{z}$." To be correct, in our training, the objective is to make $E_{\boldsymbol{z}} [p_{\boldsymbol{\theta}} ( \boldsymbol{x} | y, \boldsymbol{z})]$  approximate  the data distribution. Clearly, our objective does not necessarily require $p_{\boldsymbol{\theta}} ( \boldsymbol{x} | y, \boldsymbol{z})$ for every single $\boldsymbol{z}$ to approximate the data distribution. This is certainly unnecessary (although sufficient), and in practice, it is never working well. Thus, we never intend to make  $p_{\boldsymbol{\theta}} ( \boldsymbol{x} | y, \boldsymbol{z})$ for every single $\boldsymbol{z}$  approximate the data distribution.
> > > > >
> > > > >
> > > > > In particular, to fully understand the significant performance improvement (up to 200\%),  the training and inference should be jointly considered together as a whole. Specifically, to achieve the best result in this paper, in the inference, we do:
> > > > > \begin{align}
> > > > > \hat y = \arg \min_{y} E_{\boldsymbol{z}} \left[ p_{\boldsymbol{\theta^*}}(\boldsymbol{x} \mid y, \boldsymbol{z}) \right] = \arg \min_{y} \int_{\boldsymbol{z}}  p_{\boldsymbol{\theta^*}} \left(\boldsymbol{x} \mid y, \boldsymbol{z} \right) r(\boldsymbol{z}) d\boldsymbol{z}
> > > > > \end{align}
> > > > > or, one can do $\hat y = \arg \min_{y} E_{\boldsymbol{z}} [E_{\boldsymbol{\theta^*}} ( \boldsymbol{x} | y, \boldsymbol{z})] =  \arg \min_{y}\int_{\boldsymbol{z}} E_{\boldsymbol{\theta^*}} \left(\boldsymbol{x} \mid y, \boldsymbol{z} \right) r(\boldsymbol{z}) d\boldsymbol{z}$. Taking this inference method into account, in the training stage, we train the model by $\boldsymbol{\theta^*} = \arg \min_{\boldsymbol{\theta}} E_{\boldsymbol{z}} [p_{\boldsymbol{\theta}} ( \boldsymbol{x} | y, \boldsymbol{z})]$. That is, the model is trained to minimize $E_{\boldsymbol{z}}[p_{\boldsymbol{\theta}} ( \boldsymbol{x} | y, \boldsymbol{z})]$, which is used for inference: $\hat y=\arg \min_{y} E_{\boldsymbol{z}} \left[ p_{\boldsymbol{\theta^*}}(\boldsymbol{x} \mid y, \boldsymbol{z}) \right]$.
> > > > >
> > > > >
> > > > > Now in light of the above clarifications, we respond to the Reviewer's comments.
> > > > >
> > > > >
> > > > > The Reviewer commented: ``If the authors agree with me on this, it follows, ... the model is not really trying to approximate the real $p_{\boldsymbol{\theta}} ( \boldsymbol{x} | y, \boldsymbol{z})$."
> > > > >
> > > > >
> > > > > The activation noise parameter $\boldsymbol{z}$ which is what we introduced to the neural network always contributes to the training because it is performing the sampling of the expectation $E_{\boldsymbol{z}} \left[ p_{\boldsymbol{\theta}}(\boldsymbol{x} \mid y, \boldsymbol{z}) \right]$. In particular, because $\boldsymbol{z}$ is the sampler in the optimization problem in training, this sampler's distribution $q(\boldsymbol{z})$ must be optimized to be $q^*(\boldsymbol{z})$. Also, in Theorem 1, we mathematically showed that the optimal distribution $r^*(\boldsymbol{z})$ for inference must be the same, $r^*(\boldsymbol{z})=q^*(\boldsymbol{z})$, which is one of the major contributions in this paper.
> > > > >
> > > > >
> > > > > The Reviewer also commented: ``The model training ... I am not sure, why the authors mention inference here."
> > > > >
> > > > > We mentioned the inference, because joint understanding of the training and inference is essential to correctly explain such significant performance improvement as we discussed above.
> > > > >
> > > > > This clarifies the misunderstanding and answers the corresponding comments.
> > > > >
> > > > > Note: Due to some technical issues at OpenReview, we had to use $E[ \cdot]$ to denote the expectation.

---

> > > > > > ### Comment · Reviewer_qks6 · 2022-12-08
> > > > > > **One brief question**
> > > > > >
> > > > > > I agree that it is a sensible optimisation target to make the expectation of distribution $E_z[p_\theta(x|y,z)] = p_\theta(x|y)$ approximate the data distribution $p_D(x|y)$. This is a latent variable model with $z$ being the latent variable.
> > > > > > If this is what the authors are attempting to do, the logarithm in equation 4 should be outside the expectation, right?
> > > > > > This would amount to a maximum likelihood objective, i.e., optimise the kl divergence between $p_D(x|y)$ and $p_\theta(x|y) = E_z[p_\theta(x|y,z)]$.
> > > > > > I believe having the logarithm inside the expectation would only optimise a bound.
> > > > > > Do you agree with this?

---

> > > > > > > ### Author Response · Authors · 2022-12-09
> > > > > > > **Answer to the new question**
> > > > > > >
> > > > > > > We thank the Reviewer for the reply. We believe that the Reviewer's confusion about the difference between the probability (prob.) itself and its expectation has been addressed.
> > > > > > >
> > > > > > > To answer the Reviewer's new question, we now need to be more precise, rather than being mostly "intuitive" for simplicity. In fact, it is not perfectly precise  to say that our objective is to make $E_{\boldsymbol{z}} \left[ p_{\boldsymbol{\theta}}(\boldsymbol{x} \mid y, \boldsymbol{z}) \right]$ approximate the data distribution. Although we indeed also said so in our previous answer, it was  only for intuitive purposes, especially to help the Reviewer clarify the difference between prob. and its expectation, because the particular expression was originally from the Reviewer's very first comment: "...This objective essentially tries to make $p_{\boldsymbol{\theta}}(\boldsymbol{x} \mid y, \boldsymbol{z})$ approximate $p_D(\boldsymbol{x} \mid y)$... for each $\boldsymbol{z}$". In order to focus only on helping the Reviewer distinguish the difference between prob. and its expectation and in order not to distract the Reviewer for another issue too much, we also used such wordings. In fact, because the expression was not perfectly precise, in our previous answer, we 'intentionally' used more intuitive terminology of "data distribution" rather than  the mathematical notation $p_D(\boldsymbol{x} \mid y)$.
> > > > > > >
> > > > > > > Now that the Reviewer understands the difference between prob. and its expectation, in the following, we can make our discussions more rigorous. Precisely speaking, as clearly described in Section 3.1 and Eq. (4) in our paper, the objective of training is to maximize the log likelihood $\log p_{\boldsymbol{\theta}}(\boldsymbol{x} \mid y)$ by approximating it with $E_{\boldsymbol{z}} \left[ \log p_{\boldsymbol{\theta}}(\boldsymbol{x} \mid y, \boldsymbol{z}) \right]$ based on:
> > > > > > > \begin{equation}
> > > > > > > \log p_{\boldsymbol{\theta}}(\boldsymbol{x} \mid y)
> > > > > > > = A_{\boldsymbol{\theta}}(\boldsymbol{x}, y) = E_{\boldsymbol{z}}\left[A_{\boldsymbol{\theta}}(\boldsymbol{x} ,y \mid \boldsymbol{z}) \right]
> > > > > > > = E_{\boldsymbol{z}} \left[ \log p_{\boldsymbol{\theta}}(\boldsymbol{x} \mid y, \boldsymbol{z}) \right]
> > > > > > > \end{equation}
> > > > > > > where $A_{\boldsymbol{\theta}}(\boldsymbol{x}, y):=\log p_{\boldsymbol{\theta}}(\boldsymbol{x} \mid y)$ represents the log likelihood function. Note that no bounding techniques are used here.
> > > > > > >
> > > > > > >
> > > > > > > Regarding the Reviewer's new question, it is unclear what the Reviewer meant by "latent variable." If it meant the mathematical random vector $\boldsymbol{z}$ that is used in the above equation (in order to condition '$\log p_{\boldsymbol{\theta}}(\boldsymbol{x} \mid y)$' itself), we are fine with the naming. However, if what the Reviewer meant was the  latent variable that is commonly used in generative models (typically, in the form of $p_{\boldsymbol{\theta}}(\boldsymbol{x} \mid y) = E_{\boldsymbol{z}} \left[  p_{\boldsymbol{\theta}}(\boldsymbol{x} \mid y, \boldsymbol{z}) \right]$), it is completely different.
> > > > > > >
> > > > > > > Most of all, the definistions of our $\boldsymbol{z}$ and the commonly-used latent variable $\boldsymbol{z}$ are fundamentally different; they are apples and oranges. Taking VAEs as an example, the commonly-used latent variable $\boldsymbol{z}$ serves as the variable (in the latent space) that is supplied (as the input) to the decoder which reconstructs the input. When $\boldsymbol{z}$ means such latent variable, it is infeasible to directly compute $E_{\boldsymbol{z}} \left[  p_{\boldsymbol{\theta}}(\boldsymbol{x} \mid y, \boldsymbol{z}) \right]$ due to the curse of dimensionality. To make it feasible, in VAEs, the encoder part is introduced along with many techniques such as importance sampling, ELBO, reparameterization, etc.
> > > > > > >
> > > > > > >
> > > > > > >
> > > > > > > Our random vector $\boldsymbol{z}$ is completely different. Our $\boldsymbol{z}$ is the noise, specifically the activation noises at all neurons which are collectively denoted by a vector. This is clarified from the very beginning of the paper in Eq. (1) on page 1 and throughout the entire paper. For this fundamental difference and to avoid any confusion, in our paper, we intentionally did not call our $\boldsymbol{z}$ a latent variable; we always called it an activation noise. Again, we invite the Reviewer to carefully read the entire Section 3.1, which discussed how our activation noise $\boldsymbol{z}$ relates to dropout, loss regularization, and data augmentation.

---

> > > > > > > > ### Comment · Reviewer_qks6 · 2022-12-09
> > > > > > > > **The euqation is unfortunately incorrect**
> > > > > > > >
> > > > > > > > Thank you explicitly giving the rigorous equation on which your training loss is based:
> > > > > > > >
> > > > > > > > "
> > > > > > > >
> > > > > > > > $
> > > > > > > > \log p_{\boldsymbol{\theta}}(\boldsymbol{x} \mid y)
> > > > > > > > = A_{\boldsymbol{\theta}}(\boldsymbol{x}, y) = E_{\boldsymbol{z}}\left[A_{\boldsymbol{\theta}}(\boldsymbol{x} ,y \mid \boldsymbol{z}) \right]
> > > > > > > > = E_{\boldsymbol{z}} \left[ \log p_{\boldsymbol{\theta}}(\boldsymbol{x} \mid y, \boldsymbol{z}) \right]
> > > > > > > > $
> > > > > > > >
> > > > > > > > "
> > > > > > > >
> > > > > > > > The equation is incorrect.
> > > > > > > > If we interpret $p_{\boldsymbol{\theta}}(\boldsymbol{x} \mid y)$ and $p_{\boldsymbol{\theta}}(\boldsymbol{x} \mid y, \boldsymbol{z})$ as probability distributions, then
> > > > > > > >
> > > > > > > > $
> > > > > > > > \log p_{\boldsymbol{\theta}}(\boldsymbol{x} \mid y)
> > > > > > > > \neq E_{\boldsymbol{z}} \left[ \log p_{\boldsymbol{\theta}}(\boldsymbol{x} \mid y, \boldsymbol{z}) \right]
> > > > > > > > $.
> > > > > > > >
> > > > > > > >
> > > > > > > > As a matter of fact
> > > > > > > > $
> > > > > > > > p_{\boldsymbol{\theta}}(\boldsymbol{x} \mid y)
> > > > > > > > = E_{\boldsymbol{z}} \left[p_{\boldsymbol{\theta}}(\boldsymbol{x} \mid y, \boldsymbol{z}) \right]
> > > > > > > > $,
> > > > > > > > meaning
> > > > > > > >
> > > > > > > > $
> > > > > > > > \log p_{\boldsymbol{\theta}}(\boldsymbol{x} \mid y)
> > > > > > > > = \log E_{\boldsymbol{z}} \left[p_{\boldsymbol{\theta}}(\boldsymbol{x} \mid y, \boldsymbol{z}) \right]
> > > > > > > > $,
> > > > > > > > which is clearly different.

---

> > > > > > > > > ### Author Response · Authors · 2022-12-09
> > > > > > > > > **Thank you for your comment on the interpretation of the objective function.**
> > > > > > > > >
> > > > > > > > > We express our deepest sense of gratitude and appreciate your attitude and your comments: from the first comment all the way down to this one. Undoubtedly, your comments have been of immense value to us.
> > > > > > > > >
> > > > > > > > > Indeed, we should acknowledge that actually, the Reviewer is correct that our objective function should be interpreted as a bound of log likelihood, which is very common in likelihood-based generative models including VAEs and diffusion models. We also would like to note that, regardless of the interpretation, all our subsequent analysis (including Theorem 1 and its proof) and simulations results (including the insights) are still valid, requiring no changes.

---

### Decision · Program_Chairs · 2023-01-20

**Decision:**

Reject

**Justification For Why Not Higher Score:**

We had concerns both about the clarity of writing, correctness, formalism, as well as the extent and thoroughness of the experimental setup.

**Justification For Why Not Lower Score:**

N/A

**Metareview: Summary, Strengths And Weaknesses:**

The paper concerns EBMs trained for classification, particularly the behavior of noise added to the classification during training and inference. The claim is that (under certain conditions), adding the same noise at training and inference time is "optimal". There are some corroborating experiments (Fig 3) that non-zero noise, added both at training and inference performance better. Unfortunately, the paper is not ready as is for publication at ICLR. There were several concerns (see for details, e.g., the thread with Reviewer qks6) around the notation, formality and correctness of the main theorem; there were also concerns around how to interpret the claim that "dropout is a special case" of the main theorem, given that the latter is a multiplicative form of noise. Finally, there were also concerns around the experimental setup: in particular, the authors only compare the simplest baseline EBM, and consider different amounts of noise at train/inference time --- they do not compare with other baselines, e.g., different generative models, different regularization strategies (other than the vanilla EBM training), score-based classifiers, etc.